# On Correctness of Automatic Differentiation for Non-Differentiable Functions

**Wonyeol Lee**[†]      **Hangyeol Yu**[†]      **Xavier Rival**[‡]      **Hongseok Yang**[†]

[†]School of Computer Science, KAIST, South Korea
[‡]INRIA Paris, Département d'Informatique of ENS, and CNRS/PSL University, France
{wonyeol.lee.cs, yhk1344}@gmail.com   rival@di.ens.fr   hongseok.yang@kaist.ac.kr

## Abstract

Differentiation lies at the core of many machine-learning algorithms, and is well-supported by popular autodiff systems, such as TensorFlow and PyTorch. Originally, these systems have been developed to compute derivatives of differentiable functions, but in practice, they are commonly applied to functions with non-differentiabilities. For instance, neural networks using ReLU define non-differentiable functions in general, but the gradients of losses involving those functions are computed using autodiff systems in practice. This status quo raises a natural question: are autodiff systems correct in any formal sense when they are applied to such non-differentiable functions? In this paper, we provide a positive answer to this question. Using counterexamples, we first point out flaws in often-used informal arguments, such as: non-differentiabilities arising in deep learning do not cause any issues because they form a measure-zero set. We then investigate a class of functions, called PAP functions, that includes nearly all (possibly non-differentiable) functions in deep learning nowadays. For these PAP functions, we propose a new type of derivatives, called intensional derivatives, and prove that these derivatives always exist and coincide with standard derivatives for almost all inputs. We also show that these intensional derivatives are what most autodiff systems compute or try to compute essentially. In this way, we formally establish the correctness of autodiff systems applied to non-differentiable functions.

## 1   Introduction

Automatic differentiation or autodiff is one of the key technologies behind the dramatic progress of deep learning in recent years [4, 26, 38, 39]. It refers to the idea of developing and using a generic tool that can differentiate any function expressed as a program in a general-purpose programming language [20, 34]. Effective autodiff systems have been developed for popular programming languages [2, 3, 5, 9, 22, 28, 33, 35, 42, 44, 45]. They have enabled the development of sophisticated models and algorithms in machine learning that, in particular, involve deep neural networks [19].

This paper is concerned with one seeming contradiction of these autodiff systems: the systems have originally been developed to compute derivatives of differentiable functions, but in practice, they are commonly applied to functions with non-differentiabilities. For instance, neural networks using ReLU define non-differentiable functions in general, but the derivatives of losses involving those functions are computed using autodiff systems in practice. This status quo raises a natural question: are autodiff systems correct in any formal sense when applied to such non-differentiable functions?

A common reaction to the question is: non-differentiabilities arising in deep learning (e.g., from ReLU) do not cause any issues because they occur rarely (i.e., they form a Lebesgue-measure-zero set). In the paper, we first show that this reaction needs to be carefully re-examined at least. Using counterexamples, we point out flaws in three often-used arguments derived from this reaction. We

then present our answer. It is also positive, but based on a class of functions that satisfy a condition called *piecewise analyticity under analytic partition* (in short, PAP). These PAP functions include nearly all (possibly non-differentiable) functions in deep learning nowadays. For these PAP functions, we propose a new type of derivatives, called *intensional derivatives*, and prove that these derivatives always exist and coincide with standard derivatives for almost all inputs. These intensional derivatives behave almost as well as, and sometimes even better than, usual derivatives for differentiable functions. For instance, they always satisfy a chain rule even if functions are non-differentiable. Using these properties of intensional derivatives, we show that the intensional derivatives are what most autodiff systems compute or try to compute essentially. In this way, we formally establish the correctness of autodiff systems that compute derivatives of non-differentiable functions.

We use $(a, b)$, $(a, b]$, $[a, b)$, and $[a, b]$ to denote intervals in $\mathbb{R}$, and $\langle a_1, \ldots, a_n \rangle$ to denote tuples. For $n \in (\mathbb{Z}_{>0} \cup \{\infty\})$, $[n]$ means the set $\{1, 2, \ldots, n\}$. We call Lebesgue measure simply by measure. The detailed statements and missing proofs of our results can be found in the appendix.

## 2  Challenges

As mentioned in the introduction, practitioners frequently apply autodiff systems to functions with non-differentiabilities, and justify these out-of-scope use cases with plausible yet heuristic arguments. In this section, we analyse these arguments. We go through three claims that are often used in the arguments implicitly, and show that although looking innocent at the outset, the claims have serious flaws; they are wrong, and we provide counterexamples.

Recall a notion of correctness for an autodiff system covering non-differentiable functions [6, 20, 24]:

**Definition 1** (Correctness of Autodiff)**.** *We say that an autodiff system is* **correct** *if the following condition holds: for every measurable function $f : \mathcal{X} \to \mathbb{R}^m$ defined on an open $\mathcal{X} \subseteq \mathbb{R}^n$ and implemented by a program, if $f$ is differentiable almost everywhere (i.e., the set of inputs making $f$ non-differentiable is contained in a measure-zero set), then for almost every $x \in \mathcal{X}$, the autodiff system applied to (the program of) $f$ and the input $x$ computes the derivative of $f$ at $x$.*

The definition permits a non-differentiable function as an input to an autodiff system, as long as its non-differentiability occurs rarely (i.e., at a measure-zero subset of the input domain). For such a function, it may be impossible to compute the correct derivative for all inputs, simply because the derivative does not exist for inputs where the function is non-differentiable. Thus, the definition just requires that the system should compute the correct derivative for *most* inputs instead (i.e., for a subset of the input domain whose complement inside the domain is contained in a measure-zero set).

Proving the correctness of an autodiff system is surprisingly difficult. Nearly every autodiff system is based on a chain rule for computing the derivative of function composition, but when the component functions are non-differentiable, designing a correct version of the rule is challenging. To help the reader see this challenge, we analyse three plausible yet flawed claims about the derivative of function composition, which are sometimes used implicitly in heuristic justifications of autodiff systems.

Let $f : \mathcal{X} \to \mathcal{Y}$ and $g : \mathcal{Y} \to \mathbb{R}^l$ be measurable functions defined over open sets $\mathcal{X} \subseteq \mathbb{R}^n$ and $\mathcal{Y} \subseteq \mathbb{R}^m$. Here is the first claim:

**Claim 1** If $f$ and $g$ are differentiable almost everywhere and continuous, then $g \circ f$ should be differentiable almost everywhere.

A rationale for the claim goes as follows. In order for $g \circ f$ to be non-differentiable at $x_0$, the function $f$ has to be non-differentiable at $x_0$, or it should map $x_0$ to a non-differentiable input to $g$ and be able to vary enough in a neighbourhood of $x_0$. The claim says that such an $x_0$ is rare (from the perspective of measure). Of course, the first case that $f$ is non-differentiable at $x_0$ occurs rarely by assumption. The second case seems to happen rarely as well, because the non-differentiable inputs to $g$ are rare and $f$ is continuous: because of continuity, if $f$ maps many $x_0$'s (i.e., all the $x_0$ in some non-measure-zero set) to those rare non-differentiable inputs of $g$, it should behave as a constant function in the neighbourhoods of most of those $x_0$'s.

The rationale has a flaw, and the claim is false. The inputs $x_0$ falling into the second case are not necessarily rare. Although $f$ is continuous, it is possible that $f$ maps many $x_0$'s to some of those rare non-differentiable inputs of $g$ without acting as a constant function in a neighbourhood of each of those $x_0$'s. The precise result is summarised in the following proposition:

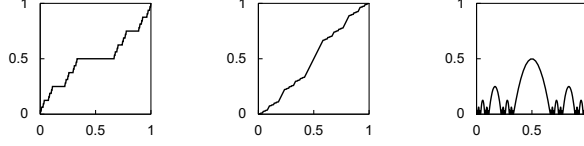

Figure 1: The graphs of $\phi_1$, $f$, and $g$ that are described in the proof of Proposition 1.

**Proposition 1.** *There exist functions $f : (0, 1) \to (0, 1)$ and $g : (0, 1) \to [0, 1]$ such that $f$ and $g$ are differentiable almost everywhere and continuous, but $g \circ f$ fails to be almost-everywhere differentiable.*

*Proof.* Before our proof, we review a generalised version of the Cantor set and Cantor function [11, 14] defined over the open interval $(0, 1)$: the $\lambda$-Cantor set $C_\lambda$ and the $\lambda$-Cantor function $\phi_\lambda$ for $\lambda \in (0, 1]$, which reduce to the original Cantor set and function when $\lambda = 1$. The $C_\lambda$ and $\phi_\lambda$ will serve as building blocks of counterexamples presented in this paper. The set $C_\lambda \subset (0, 1)$ consists of all the real numbers not removed during the following procedure: at step $k = 0$, start with a closed interval $[0, 1] \subset \mathbb{R}$; at each step $k > 0$, remove every open interval of length $\lambda/3^k$ which locates exactly at the middle of each remaining closed interval; and after all these steps, remove $0$ and $1$. Note that what remains after the step $k$ are $2^k$ closed intervals. The function $\phi_\lambda : (0, 1) \to (0, 1)$ is defined as follows, based on the construction of $C_\lambda$. At each step $k > 0$, define $\phi_\lambda$ over the $i$-th open interval to be removed as $\phi_\lambda(x) = (2i - 1)/2^k$ ($i \in [2^{k-1}]$). After all these steps, $\phi_\lambda$ is defined over some dense open subset $S$ of $[0, 1]$ (in fact, $S = (0, 1) \setminus C_1$). Since $\phi_\lambda : S \to [0, 1]$ is uniformly continuous with $S$ being dense in $[0, 1]$, it has the unique continuous extension $\phi_\lambda : [0, 1] \to [0, 1]$ [37, Exercise 4.13]. The $\lambda$-Cantor function refers to its restricted version $\phi_\lambda : (0, 1) \to (0, 1)$.

Returning to the proof, let $f$ be the inverse of the homeomorphism $F : (0, 1) \to (0, 1)$ defined by $F(x) = \frac{1}{2}(\phi_1(x) + x)$ [8, Example 2.3.1]. It is known that $f(C_{1/2}) = C_1$ [8, Example 2.3.2]. Construct $g : (0, 1) \to [0, 1]$ based on the construction of $C_1$ as follows, similarly to [18, Example 8.18]: at each step $k > 0$, define $g$ over the $i$-th open interval $(l_i, r_i)$ to be removed, as $g(y) = 2^{-k} \cdot [1 - (y - (r_i + l_i)/2)^2/((r_i - l_i)/2)^2]$ ($i \in [2^{k-1}]$); and define $g$ over $C_1$ as $g(y) = 0$. See Figure 1 for the graphs of $\phi_1$, $f$, and $g$ constructed so far. Clearly, $f$ is continuous. Also, $g$ is continuous, since the height of the parabolas defined at the $k$-th step of $g$'s construction converges to $0$ as $k \to \infty$, and $g(C_1) = \{0\}$ (see Appendix A for the details). Hence, $g \circ f$ is continuous. Note that $f$ is even Lipschitz continuous. To prove this, observe that $f$ can be constructed similarly to $\phi_{1/2}$, and repeat the proof of the Lipschitz continuity of $\phi_{1/2}$ [14].

We now show that $f$ and $g$ are differentiable almost everywhere, but $g \circ f$ is not. First, since $f$ is Lipschitz continuous, it is differentiable almost everywhere by Rademacher's theorem [41, Theorem 2.2.4]. Next, since $g$ is differentiable on $(0, 1) \setminus C_1$ by its construction, it is differentiable almost everywhere (as $C_1$ has measure 0). Lastly, $g \circ f$ is non-differentiable on $C_{1/2}$, which has measure $1/2$, due to that: the parabolas defined at the $k$-th step of $g$'s construction get sharper as $k \to \infty$; $g(C_1) = \{0\}$; and $f$ is a homeomorphism with $f(C_{1/2}) = C_1$ (see Appendix A for the details). $\square$

The second claim is about a chain rule.

**Claim 2** If $f$, $g$, and $g \circ f$ are differentiable almost everywhere and continuous, then the standard chain rule for $g \circ f$ should hold almost everywhere. In particular, if $f, g : (0, 1) \to (0, 1)$, then $(g \circ f)'(x_0) = g'(f(x_0)) \cdot f'(x_0)$ for almost all $x_0 \in (0, 1)$.

Note that all of $f$, $g$, and $g \circ f$ in the claim are assumed to be differentiable almost everywhere. The claim comes from heuristic reasoning that if we just avoid those rare non-differentiable inputs of $g \circ f$, we should be able to use the standard result for differentiation, including the chain rule.

The second claim is also wrong. The flaw in the heuristic reasoning from above is that the almost-everywhere differentiability of $f$, $g$, and $g \circ f$ does not stop $g'(f(x_0))$ from being undefined for many $x_0$'s in $(0, 1)$. This is related to the flaw in the justification for the first claim that we explained. The next proposition and its proof provide a concrete example for this phenomenon:

**Proposition 2.** *There exist functions $f, g : (0, 1) \to (0, 1)$ such that $f$, $g$, and $g \circ f$ are differentiable almost everywhere and continuous, but it is not that $g'(f(x))$ is defined for almost all $x \in (0, 1)$.*

*Proof.* Let $f(x) = 1/2$ and $g(y) = \mathrm{ReLU}(y - 1/2) + 1/2$. Then, $(g \circ f)(x) = 1/2$. Certainly, $f$, $g$, and $g \circ f$ are differentiable almost everywhere and Lipschitz continuous. But, $g$ is not differentiable at $f(x) = 1/2$ for all $x \in (0, 1)$. So, it is not that $g'(f(x))$ is defined for almost all $x \in (0, 1)$. $\qquad \square$

The third claim is a natural reaction to the failure of the second claim. It implements the strategy of making the chain rule in the second claim more permissive such that the counter argument of Proposition 2 no longer applies. The claim expresses a weaker version of the rule that allows one to set the derivatives of $f$ and $g$ to arbitrary values wherever $f$ and $g$ are not differentiable.

**Claim 3** Let $f, g : (0, 1) \to (0, 1)$. If $f$, $g$, and $g \circ f$ are differentiable almost everywhere and continuous, then there should exist $df, dg : (0, 1) \to \mathbb{R}$ such that $df(x_0) = f'(x_0)$, $dg(y_0) = g'(y_0)$, and $(g \circ f)'(x_0) = dg(f(x_0)) \cdot df(x_0)$ for almost all $x_0, y_0 \in (0, 1)$.

The functions $df$ and $dg$ in the claim are the extensions of $f'$ and $g'$ that set $df(x)$ and $dg(y)$ to arbitrary values whenever $f'(x)$ and $g'(y)$ are undefined. The chain rule in the claim is phrased in terms of these extensions $df$ and $dg$, so that it does not suffer from the problem pointed out in Proposition 2. However, this new rule is still flawed as shown in the next proposition:

**Proposition 3.** *There exist functions $f, g : (0, 1) \to (0, 1)$ such that $f$, $g$, and $g \circ f$ are differentiable almost everywhere and continuous, but for some measurable subset $A \subseteq (0, 1)$ with non-zero measure, they satisfy the following property: $f'(x) = 0$ and $(g \circ f)'(x) \neq 0$ for all $x \in A$.*

*Proof.* Consider the function $f$ in the proof of Proposition 1. Let $g$ be the 1-Cantor function $\phi_1$. Then, $g \circ f$ is the $(1/2)$-Cantor function $\phi_{1/2}$. We already showed $f$ is differentiable almost everywhere and even Lipschitz continuous. Since $g$ and $g \circ f$ are monotone on $(0, 1)$, they are differentiable almost everywhere by the monotone differentiation theorem [41, Theorem 1.6.25]; and they are clearly continuous. We now show there exists $A \subseteq C_{1/2}$ with the desired properties. Since $C_{1/2}$ has measure $1/2$, it suffices to prove $f'(x) = 0$ and $(g \circ f)'(x) = 2$ for almost all $x \in C_{1/2}$. The claim indeed holds due to the following: $f$ and $g \circ f$ are Lipschitz, so absolutely continuous; $f'(x) = 2$ and $(g \circ f)'(x) = 0$ for all $x \notin C_{1/2}$; $f'(x) \geq 0$ and $(g \circ f)'(x) \leq 2$ for all $x \in C_{1/2}$ whenever these derivatives exist; and $C_{1/2}$ has measure $1/2$. For the details, see [8, Example 2.3.2] and [14]. $\qquad \square$

The proposition implies the third claim is doomed. The claim says that $df(x_0) = f'(x_0)$ and $(g \circ f)'(x_0) = dg(f(x_0)) \cdot df(x_0)$ for almost all $x_0 \in A$. But both equalities cannot hold simultaneously: if they do, by Proposition 3, $(g \circ f)'(x_0) = dg(f(x_0)) \cdot df(x_0) = dg(f(x_0)) \cdot f'(x_0) = dg(f(x_0)) \cdot 0 = 0$, but the same proposition also entails $(g \circ f)'(x_0) \neq 0$, leading to a contradiction.

A lesson from these flawed claims is that although the notion of correctness in Definition 1 only refers to almost-everywhere differentiability, we need a condition stronger than it, which behaves better in handling function composition and gives rise to a chain rule. We describe such a condition next.

## 3 PAP Function and Intensional Derivative

Our justification of autodiff systems relies on two key concepts: *piecewise analyticity under analytic partition*, and *intensional derivative*. The first is a (strictly) stronger property about functions than almost-everywhere differentiability, and yet it is satisfied by practically all programs targeted at by existing autodiff systems, as we will show in §4. Functions with this new property, called PAP functions, have an unusual type of derivatives, called intensional derivatives, which form the second concept. Intensional derivatives of PAP functions are defined everywhere and satisfy a chain rule, while still agreeing with standard derivatives for almost all inputs. In fact, the PAP functions have not just first-order but also all higher-order intensional derivatives. In §4, we will show that most autodiff systems compute intensional derivatives when applied to functions with non-differentiabilities.

To expand the overview of the two concepts just given, we need a notion of piecewise representation:

**Definition 2** (Piecewise Representation)**.** *Let $\mathcal{X} \subseteq \mathbb{R}^n$ and $\mathcal{Y} \subseteq \mathbb{R}^m$. A set of pairs $\gamma = \{\langle A^i, f^i \rangle\}_{i \in [I]}$ is a **piecewise representation** of a function from $\mathcal{X}$ to $\mathcal{Y}$ or simply **representation** if and only if (i) $I \in \mathbb{Z}_{>0} \cup \{\infty\}$; (ii) $\{A^i\}_{i \in [I]}$ is a partition of $\mathcal{X}$; and (iii) each $f^i : \mathcal{X}^i \to \mathcal{Y}$ is a function on an open domain $\mathcal{X}^i \subseteq \mathbb{R}^n$ with $A^i \subseteq \mathcal{X}^i$. The **evaluation** of a representation $\gamma = \{\langle A^i, f^i \rangle\}_{i \in [I]}$ is the map $\langle\!\langle \gamma \rangle\!\rangle : \mathcal{X} \to \mathcal{Y}$ defined by: $\langle\!\langle \gamma \rangle\!\rangle(x) = f^i(x)$ for all $i \in [I]$ and $x \in A^i$.*

A representation $\gamma = \{\langle A^i, f^i \rangle\}_{i \in [I]}$ describes a function $f$ in terms of a finite or countable number of component functions $f^i$ and their scopes $A^i$ of application. For each input $x$, the value of $f$ at $x$ is determined by some component $f^i$, chosen based on which piece of the input partition $\{A^i\}_{i \in [I]}$, the $x$ belongs to. Our notation for the described $f$ is $\langle\!\langle \gamma \rangle\!\rangle$. Note that the domain $\mathcal{X}^i$ of each $f^i$ has to be open, and it may be larger than $A^i$, the set of inputs where $f^i$ is used. As a result, a function can be described by an infinite number of representations. For instance, ReLU has at least two representations: $\{\langle \mathbb{R}_{<0}, x \longmapsto 0 \rangle, \langle \mathbb{R}_{\geq 0}, x \longmapsto x \rangle\}$ and $\{\langle \mathbb{R}_{<0}, x \longmapsto 0 \rangle, \langle \{0\}, x \longmapsto 2x \rangle, \langle \mathbb{R}_{>0}, x \longmapsto x \rangle\}$.

Having a piecewise representation of a function, instead of just the function, has several advantages. One of them is that for many properties of functions, we can derive their piecewise variants using those representations. We apply this advantage to analyticity. Recall that a real-valued function is analytic if and only if it is infinitely differentiable and is equal to its Taylor expansion. For $\mathbb{R}^m$-valued function $g$, the analyticity means the coordinate-wise analyticity: for all $j \in [m]$, the composition $\pi_j \circ g$ of the $j$-th projection $\pi_j$ and $g$ is analytic. Analytic functions are favoured by autodiff systems, because they are infinitely differentiable and become zero only at a measure-zero set of inputs (unless they are zero everywhere or their domains are disconnected). Let $\mathcal{X} \subseteq \mathbb{R}^n$ and $\mathcal{Y} \subseteq \mathbb{R}^m$ be any sets.

**Definition 3** (Analytic Partition). *A set $A \subseteq \mathbb{R}^n$ is **analytic** if and only if for some $J, L \in \mathbb{Z}_{>0}$, there are analytic functions $g_j^+ : \mathcal{X}_j^+ \to \mathbb{R}$ and $g_l^- : \mathcal{X}_l^- \to \mathbb{R}$ over open domains $\mathcal{X}_j^+, \mathcal{X}_l^- \subseteq \mathbb{R}^n$ ($j \in [J]$, $l \in [L]$) such that $A = \{x \in \mathbb{R}^n \mid (\bigwedge_{j \in [J]} x \in \mathcal{X}_j^+ \land g_j^+(x) > 0) \land (\bigwedge_{l \in [L]} x \in \mathcal{X}_l^- \land g_l^-(x) \leq 0)\}$. A partition $\{A^i\}_{i \in [I]}$ of $\mathcal{X}$ is **analytic** if and only if all $A^i$ are analytic.*

**Definition 4** (PAP Representation). *A representation $\gamma = \{\langle A^i, f^i \rangle\}_{i \in [I]}$ from $\mathcal{X}$ to $\mathcal{Y}$ is **piecewise analytic under analytic partition** (in short, **PAP**) if and only if $\{A^i\}_{i \in [I]}$ is an analytic partition of $\mathcal{X}$ and $f^i$ is analytic over its domain $\mathcal{X}^i$ for all $i \in [I]$.*

**Definition 5** (PAP Function). *A function $f : \mathcal{X} \to \mathcal{Y}$ is **piecewise analytic under analytic partition** (in short, **PAP**) if $f = \langle\!\langle \gamma \rangle\!\rangle$ for some PAP representation $\gamma$.*

The definitions identify PAP representations and PAP functions as those built by the two-step process: we first split the input domain such that boundaries of the split regions are expressed by the zero sets of analytic functions, and next choose an appropriate analytic function for each piece of the split. Note the use of analytic functions in both steps. Thus, just like the standard analyticity, the PAP property implies almost-everywhere differentiability (Proposition 4), but not vice versa (Proposition 5).

**Proposition 4** ([47]). *All PAP functions are differentiable almost everywhere.*

*Proof.* The proof extends the one for a similar result in [47]. The key idea is to use the fact that the zero set of a non-constant analytic function over a connected open domain has measure zero [32]. To prove the proposition, we show that for each PAP function $f$, there exist countably many non-constant analytic functions $\{g_j\}_j$ over connected open domains such that if $f$ is non-differentiable at $x \in \mathcal{X}$, then $x$ belongs to the zero set of some $g_j$. For the details, see Appendix B.3. $\qquad\square$

**Proposition 5.** *There is a continuous almost-everywhere differentiable yet non-PAP function.*

*Proof.* We have two sufficient conditions for a function $f$ to be non-PAP: (i) the $k$-th order derivative of $f$ is undefined on a set of positive measure for some $k \geq 1$ (Proposition 8); and (ii) the $k$-th order derivative of $f$ is undefined on an uncountable set for some $k \geq 1$, and $f$ is defined on a subset of $\mathbb{R}$ (Appendix B.5). The following functions satisfy (i) with $k = 2$: the $\lambda$-Cantor function for all $\lambda \in (0, 1)$, $f$ in the proof of Proposition 1, and Volterra's function $V : (0, 1) \to \mathbb{R}$ [18, Example 8.35]. The following functions satisfy (ii) with $k = 1$: the 1-Cantor function and $g$ in the proof of Proposition 1. All these functions are known (or already proven above) to be continuous and almost-everywhere differentiable. Hence, all of them are desired functions. $\qquad\square$

We now define intensional derivatives of PAP representations and PAP functions. Let $\gamma = \{\langle A^i, f^i \rangle\}_{i \in [I]}$ be a PAP representation from $\mathcal{X}$ to $\mathcal{Y}$ for some $\mathcal{X} \subseteq \mathbb{R}^n$ and $\mathcal{Y} \subseteq \mathbb{R}^m$.

**Definition 6** (Intensional Derivative of PAP Representation). *An **intensional derivative** of $\gamma$ is the set $D\gamma = \{\langle A^i, Df^i \rangle\}_{i \in [I]}$, where $Df^i$ is the standard derivative of $f^i$ viewed as a function from the domain $\mathcal{X}^i$ of $f^i$ to $\mathbb{R}^{m \times n}$.*

**Proposition 6.** *The intensional derivative $D\gamma$ is a PAP representation from $\mathcal{X}$ to $\mathbb{R}^{m \times n}$.*

*Proof.* Nearly all the requirements for $D\gamma$ to be a PAP representation directly follow from the fact that $\gamma$ is PAP. The only exception is the analyticity of $Df^i$. There we use the fact that the operation of taking a (standard) partial derivative of a function preserves analyticity [25, Proposition 2.2.3]. $\qquad\square$

**Definition 7** (First-Order Intensional Derivative). *Let $f : \mathcal{X} \to \mathcal{Y}$ be a PAP function. Define $\partial_\bullet f$ to be the following set of functions: $\partial_\bullet f = \{\langle\!\langle D\gamma \rangle\!\rangle \mid \gamma \text{ is a PAP representation of } f\}$. Each $df \in \partial_\bullet f$ is called a (**first-order**) **intensional derivative** of $f$.*

By Proposition 6, only PAP functions live in $\partial_\bullet f$. Thus, we can also take intensional derivatives of functions in $\partial_\bullet f$. We push this observation further and define higher-order intensional derivatives:

**Definition 8** (Higher-Order Intensional Derivative). *Let $f : \mathcal{X} \to \mathcal{Y}$ be a PAP function. For each $k \in \mathbb{Z}_{\geq 0}$, inductively define the set $\partial_\bullet^k f$ of functions by: $\partial_\bullet^0 f = \{f\}$, and $\partial_\bullet^k f = \{df^k \in \partial_\bullet(df^{k-1}) \mid df^{k-1} \in \partial_\bullet^{k-1} f\}$ for $k \geq 1$. Each $df^k \in \partial_\bullet^k f$ is called a $k$-**th order intensional derivative** of $f$.*

A function $f : \mathcal{X} \to \mathcal{Y}$ may have zero, one, or more than one intensional derivatives. Having at least one intensional derivative corresponds to $f$ being differentiable in the standard sense. The next propositions show that every PAP function is infinitely differentiable in the new sense (Proposition 7), and that these standard and new notions of differentiability coincide if we ignore a measure-zero set of inputs (Proposition 8). Let $f : \mathcal{X} \to \mathcal{Y}$ be a PAP function for some $\mathcal{X} \subseteq \mathbb{R}^n$ and $\mathcal{Y} \subseteq \mathbb{R}^m$.

**Proposition 7.** *For all $k \in \mathbb{Z}_{\geq 0}$, the set $\partial_\bullet^k f$ of the $k$-th order intensional derivatives is not empty. Furthermore, its elements are again PAP functions from $\mathcal{X}$ to $\mathbb{R}^{m \times n^k}$.*

*Proof.* The proof is by induction on $k$. First, for the case $k = 0$, $\partial_\bullet^k f$ is a singleton set and its unique element $f$ is PAP. Hence, the proposition holds. Next, consider the case $k > 0$. By induction hypothesis, $\partial_\bullet^{k-1} f$ is not empty and consists of PAP functions only. Thus, the set $\{\gamma \mid \gamma \text{ is a PAP representation of } g \text{ for some } g \in \partial_\bullet^{k-1} f\}$ is not empty. This and Proposition 6 imply that $\partial_\bullet^k f$ (obtained by applying $\langle\!\langle D- \rangle\!\rangle$ on the above set) is nonempty and contains only PAP functions. $\square$

**Proposition 8.** *For all $k \in \mathbb{Z}_{\geq 0}$, $f$ has the $k$-th order standard derivative almost everywhere, and this derivative agrees with any $k$-th order intensional derivative $df^k \in \partial_\bullet^k f$ almost everywhere.*

In the proposition, we view the $k$-th order standard derivative of $f$ as a function of type $\mathcal{X} \to \mathbb{R}^{m \times n^k}$. For instance, when $k = 1$, the derivative maps an input to the Jacobian matrix of $f$.

*Proof of Proposition 8.* The first claim is proven similarly to Proposition 4, except that we additionally use the following: an analytic function is infinitely differentiable. As in Proposition 4, we prove a stronger statement: there exist countably many non-constant analytic functions $\{g_j\}_j$ over connected open domains such that for all $k$, if the $k$-th order standard derivative of $f$ is not defined at $x \in \mathcal{X}$, then $x$ is in the zero set of some $g_j$. Next, consider the second claim. Its current form is not strong enough to enable inductive proofs. We instead prove a stronger statement by induction on $k$: for each $df^k \in \partial_\bullet^k f$, there exist countably many non-constant analytic functions $\{h_l\}_l$ over connected open domains such that the $k$-th order standard derivative of $f$ is well-defined, and agrees with $df^k$, at all those inputs not in the zero sets of $\{h_l\}_l$. For the details, see Appendices B.3 and B.4. $\square$

Intensional derivatives behave better than standard derivatives. First, the intensional derivatives of a PAP function $f$ are total functions, i.e., functions defined for all inputs (Proposition 7). Contrast this with the fact that the standard derivative of $f$ is a partial function in general. The intensional derivatives can be understood as extensions of this standard derivative of $f$ to those problematic inputs that make $f$ non-differentiable (Proposition 8). The totality simplifies the reasoning about intensional derivatives. Second, the intensional derivatives satisfy a chain rule for all PAP functions (Proposition 10). Let $f : \mathcal{X} \to \mathcal{Y}$ and $g : \mathcal{Y} \to \mathbb{R}^l$ be PAP functions for some $\mathcal{X} \subseteq \mathbb{R}^n$ and $\mathcal{Y} \subseteq \mathbb{R}^m$.

**Proposition 9.** *$g \circ f$ is a PAP function.*

*Proof.* Since $f$ and $g$ are PAP, they have PAP representations $\gamma_f = \{\langle A^i, f^i \rangle\}_{i \in [I]}$ and $\gamma_g = \{\langle B^j, g^j \rangle\}_{j \in [J]}$. Define their composition as follows: $\gamma_g \circ \gamma_f = \{\langle C^{\langle i,j \rangle}, g^j \circ f^i \rangle\}_{\langle i,j \rangle \in [I] \times [J]}$ where $C^{\langle i,j \rangle} = \{x \in \mathcal{X} \mid x \in A^i \wedge f^i(x) \in B^j\}$. Then, $\gamma_g \circ \gamma_f$ is a representation of $g \circ f$. Also, it is PAP as the composition of analytic functions is analytic [25, Proposition 2.2.8]. Thus, $g \circ f$ is PAP. $\square$

**Proposition 10** (Chain Rule for Intensional Derivatives). *Let $df : \mathcal{X} \to \mathbb{R}^{m \times n}$ and $dg : \mathcal{Y} \to \mathbb{R}^{l \times m}$ be intensional derivatives of $f$ and $g$ (i.e., $df \in \partial_\bullet f$ and $dg \in \partial_\bullet g$). Let $h = g \circ f$. Then, the following function $dh : \mathcal{X} \to \mathbb{R}^{l \times n}$ is an intensional derivative of $h$ (i.e., $dh \in \partial_\bullet h$): $dh(x) = dg(f(x)) \cdot df(x)$ for all $x \in \mathcal{X}$, where $dg(f(x)) \cdot df(x)$ is the multiplication of matrices $dg(f(x))$ and $df(x)$.*

*Proof.* By the definition of intensional derivative, $df = \langle\!\langle D\gamma_f \rangle\!\rangle$ and $dg = \langle\!\langle D\gamma_g \rangle\!\rangle$ for some PAP representations $\gamma_f = \{\langle A^i, f^i \rangle\}_{i \in [I]}$ and $\gamma_g = \{\langle B^j, g^j \rangle\}_{j \in [J]}$ of $f$ and $g$. Let $\gamma_g \circ \gamma_f$ be the composed representation defined in the proof of Proposition 9. Then, $\gamma_g \circ \gamma_f$ is a PAP representation

of $g \circ f$. So, $\langle\!\langle D(\gamma_g \circ \gamma_f) \rangle\!\rangle$ is an intensional derivative of $g \circ f$. Also, for all $x \in \mathcal{X}$, if we let $\langle i, j \rangle \in [I] \times [J]$ with $x \in A^i$ and $f^i(x) \in B^j$, then $\langle\!\langle D(\gamma_g \circ \gamma_f) \rangle\!\rangle(x) = D(g^j \circ f^i)(x) = D(g^j)(f^i(x)) \cdot D(f^i)(x) = \langle\!\langle D\gamma_g \rangle\!\rangle(f(x)) \cdot \langle\!\langle D\gamma_f \rangle\!\rangle(x)$. The first and last equalities follow from the definition of intensional derivative. The second equality uses the chain rule for standard derivatives: the rule holds here because $f^i$ and $g^j$ are analytic in neighbourhoods of $x$ and $f^i(x)$, respectively. $\quad\square$

We next use these properties to show that existing autodiff systems compute intensional derivatives.

## 4  Correctness of Autodiff Systems

Consider a simple programming language that assumes real-valued input variables $x_1, \ldots, x_N$ and has the following syntax for programs: $e ::= c \mid x_i \mid \bar{\mathtt{f}}(e_1, \ldots, e_n) \mid \mathtt{if}\ (e_1 > 0)\ e_2\ e_3$.

A program $e$ in the language describes a real-valued computation. It is a real number $c$, an input variable $x_i$, the application of a primitive function $\bar{\mathtt{f}}$ to arguments $e_1, \ldots, e_n$, or a conditional expression. In the third case, the applied function is required to be a PAP function of the right type (i.e., $\mathbb{R}^n \to \mathbb{R}$ in this case). We remark that practically all primitive functions supported by autodiff systems are indeed PAP; we are unaware of any non-PAP primitive function used in practice. If this requirement is met, all programs $e$ mean PAP functions of type $\mathbb{R}^N \to \mathbb{R}$. More precisely, we interpret each program $e$ as a function $[\![e]\!] : \mathbb{R}^N \to \mathbb{R}$ inductively, as shown below: for all $v \in \mathbb{R}^N$,

$$[\![c]\!]v = c, \qquad [\![x_i]\!]v = v_i, \qquad [\![\bar{\mathtt{f}}(e_1, \ldots, e_n)]\!]v = \mathtt{f}([\![e_1]\!]v, \ldots, [\![e_n]\!]v),$$

$$[\![\mathtt{if}\ (e_1 > 0)\ e_2\ e_3]\!]v = \mathrm{if}\ ([\![e_1]\!]v > 0)\ \text{then}\ [\![e_2]\!]v\ \text{else}\ [\![e_3]\!]v.$$

where $v_i$ is the $i$-th component of the vector $v$, and $\mathtt{f} : \mathbb{R}^n \to \mathbb{R}$ is the PAP function denoted by the function symbol $\bar{\mathtt{f}}$ in the language. Then, the defined $[\![e]\!]$ is always PAP.

**Proposition 11.** *For every program $e$, its denotation $[\![e]\!]$ is a PAP function from $\mathbb{R}^N$ to $\mathbb{R}$.*

We show that if an autodiff system for the language satisfies two requirements to be described shortly, it essentially computes an intensional derivative, even when the input function is not differentiable. The first requirement is that for every primitive operation $\bar{\mathtt{f}}$ of type $\mathbb{R}^n \to \mathbb{R}$, the system should come with a function $\widetilde{D}\mathtt{f} : \mathbb{R}^n \to \mathbb{R}^{1 \times n}$ that satisfies $\widetilde{D}\mathtt{f} \in \partial_\bullet \mathtt{f}$ and serves as a "derivative" of $\mathtt{f}$. To describe the second requirement, we inductively define the function $[\![e]\!]^\nabla : \mathbb{R}^N \to \mathbb{R}^{1 \times N}$, formalising symbolic differentiation of a program $e$, as follows: for all $v \in \mathbb{R}^n$,

$$[\![c]\!]^\nabla v = \vec{0}_{1 \times N}, \quad [\![\bar{\mathtt{f}}(e_1, \ldots, e_n)]\!]^\nabla v = (\widetilde{D}\mathtt{f})([\![e_1]\!]v, \ldots, [\![e_n]\!]v) \cdot [[\![e_1]\!]^\nabla v; \ldots; [\![e_n]\!]^\nabla v],$$

$$[\![x_i]\!]^\nabla v = [\vec{0}_{(i-1) \times 1}; \vec{1}_{1 \times 1}; \vec{0}_{(N-i) \times 1}]^\top, \quad [\![\mathtt{if}\ (e_1 > 0)\ e_2\ e_3]\!]^\nabla v = \mathrm{if}\ ([\![e_1]\!]v > 0)\ \text{then}\ [\![e_2]\!]^\nabla v\ \text{else}\ [\![e_3]\!]^\nabla v.$$

Here $\vec{c}_{i \times j}$ denotes the $i \times j$ matrix containing only the constant $c$, and the RHS of the second equation means the multiplication of the $1 \times n$ matrix $(\widetilde{D}\mathtt{f})([\![e_1]\!]v, \ldots, [\![e_n]\!]v)$ and the $n \times N$ matrix $[[\![e_1]\!]^\nabla v; \ldots; [\![e_n]\!]^\nabla v]$ that is constructed by concatenating $n$ matrices $[\![e_1]\!]^\nabla v, \ldots, [\![e_n]\!]^\nabla v$ of size $1 \times N$. The definition of $[\![e]\!]^\nabla v$ computes a "Jacobian" matrix of $[\![e]\!]$ (which is a $1 \times N$ matrix in this case) in the usual way of symbolic differentiation: run the program $e$ under the input $v$, and calculate the "Jacobian" matrix of the encountered operations using the chain rule with the "derivatives" $\widetilde{D}\mathtt{f}$ of those operations $\mathtt{f}$ [1, 4, 23, 46]. Our second requirement is that given a program $e$ and an input $v \in \mathbb{R}^N$, if the system performs forward-mode (or reverse-mode) autodiff with a tangent vector $w \in \mathbb{R}^N$ (or a cotangent vector $u \in \mathbb{R}$), it should output the Jacobian-vector product $([\![e]\!]^\nabla v) \cdot w \in \mathbb{R}$ (or the vector-Jacobian product $u^\top \cdot ([\![e]\!]^\nabla v) \in \mathbb{R}^N$) [20, 27, 31]. Intuitively, the second requirement says that the output of an autodiff system should coincide with that of symbolic differentiation. Note that the requirement does not fix the choice of an autodiff algorithm, and has two separate conditions for forward-mode and reverse-mode algorithms. Our next results show that these requirements ensure the correctness of an autodiff system.

**Theorem 12.** *If $\widetilde{D}\mathtt{f} \in \partial_\bullet \mathtt{f}$ for all primitive functions $\bar{\mathtt{f}}$, then $[\![e]\!]^\nabla \in \partial_\bullet [\![e]\!]$ for all programs $e$.*

**Corollary 13.** *Assume that an autodiff system for the language in this section satisfies the two requirements. Then, for each program $e$, there exists an intensional derivative $df$ of $[\![e]\!]$ such that if the system performs forward-mode (or reverse-mode) autodiff with a tangent vector $w \in \mathbb{R}^N$ (or a cotangent vector $u \in \mathbb{R}$), it computes the Jacobian-vector product $df(v) \cdot w \in \mathbb{R}$ (or the vector-Jacobian product $u^\top \cdot df(v) \in \mathbb{R}^N$) for every input $v \in \mathbb{R}^N$. Furthermore, the computed entity is the corresponding Jacobian-vector product (or vector-Jacobian product) with the standard derivative of $[\![e]\!]$ for almost all inputs $v \in \mathbb{R}^N$.*

**Remark 1** (Intensional Derivatives in Practice). We briefly discuss whether the first requirement mentioned earlier is indeed met by autodiff systems. TensorFlow [2] and PyTorch [33], two popular autodiff systems, support a wide range of primitive functions that are not differentiable for some inputs. A well-known example is `relu` : $\mathbb{R} \to \mathbb{R}$ (i.e., the map $x \longmapsto \max(0, x)$). This function is not differentiable at $0$, but when these autodiff systems are applied to differentiate `relu` at $0$, they return $0$ instead of an error (i.e., $(\widetilde{D}\text{relu})(0) = 0$). It means that the systems compute the intensional derivative $\langle\!\langle D\gamma \rangle\!\rangle$ of `relu` for $\gamma = \{\langle \mathbb{R}_{>0}, x \longmapsto x \rangle, \langle \mathbb{R}_{\leq 0}, x \longmapsto 0 \rangle\}$. Thus, they fulfil the first requirement on `relu`. Note that setting the derivative at the input $0$ to any $c \in \mathbb{R}$ is an acceptable option; it can be justified by a different PAP representation $\gamma'$ of `relu` where $\gamma' = \{\langle \mathbb{R}_{>0}, x \longmapsto x \rangle, \langle \{0\}, x \longmapsto c \cdot x \rangle, \langle \mathbb{R}_{<0}, x \longmapsto 0 \rangle\}$. Another example is `reciprocal_no_nan` from TensorFlow. It means the map $f(x) = 1/x$ if $x \neq 0$, and $0$ if $x = 0$. Differentiating this map by TensorFlow gives the function $(\widetilde{D}f)(x) = -1/x^2$ if $x \neq 0$, and $0$ if $x = 0$. The $\widetilde{D}f$ is an intensional derivative of $f$, because $\gamma = \{\langle \mathbb{R} \setminus \{0\}, x \longmapsto 1/x \rangle, \langle \{0\}, x \longmapsto 0 \rangle\}$ is a PAP representation of $f$ and $\langle\!\langle D\gamma \rangle\!\rangle$ coincides with $\widetilde{D}f$. Hence, TensorFlow meets the first requirement on `reciprocal_no_nan`.

However, for some functions, TensorFlow and PyTorch behave more conservatively than our theory of PAP functions permits. A good example is `sqrt`. The domain of `sqrt` is $\mathbb{R}_{\geq 0}$, but differentiating `sqrt` at $0$ using these autodiff systems returns `+inf`, which means $\infty$ (i.e., $(\widetilde{D}\text{sqrt})(0) = \infty$). Thus, the first requirement on `sqrt` is violated by these systems. One consequence of this is that when the systems are applied to differentiate `sqrt(mult(x,0))`, they end up with computing `0*(+inf)` that evaluates to `NaN` (meaning an undefined number in floating point arithmetic) for all $x \in \mathbb{R}$. Note that `sqrt(mult(x,0))` is constantly zero and has zero as its derivative. This conservative behaviour could have been avoided if TensorFlow and PyTorch had recognised that `sqrt` is a PAP function and had used its intensional derivative instead. For instance, the function $(d\text{sqrt})(x) = 1/(2\sqrt{x})$ if $x > 0$, and $0$ if $x = 0$, is an intensional derivative of `sqrt`, since $\gamma = \{\langle \mathbb{R}_{>0}, x \longmapsto \sqrt{x} \rangle, \langle \{0\}, x \longmapsto 0 \rangle\}$ is a PAP representation of `sqrt`. If TensorFlow and PyTorch used $d\text{sqrt}$ as a derivative of `sqrt`, they would be able to differentiate `sqrt(mult(x,0))` correctly for all $x \in \mathbb{R}$.

## 5 Related Work and Discussion

Autodiff has a long history with a large body of literature [4, 20, 34]. Its community has been aware of some issues with non-differentiable functions [20, Chapter 14]. These issues have become ever more important, as autodiff has been increasingly applied to a variety of non-differentiable functions, including sophisticated linear algebra functions [24, 27, 40]. In this paper, we investigate the issues in a more systematic and rigorous way, by presenting non-trivial concrete counterexamples that illuminate subtleties of non-differentiable functions in autodiff, and also proposing intensional derivatives, a new notion of derivatives, that enable us to formally prove the correctness of, and better understand the behaviour of, autodiff systems applied to non-differentiable functions.

Clarke subdifferential [12, 13] and related notions have been favourite tools for studying the use of non-differentiable functions in the context of autodiff and stochastic gradient descent [6, 15, 21, 24, 29]. Let $f$ be an input program (or function) to be differentiated. Assume that $f$ is locally Lipschitz continuous but possibly non-differentiable at some inputs. Kakade and Lee [24] proposed a linear-time algorithm that correctly computes a *subdifferential* of $f$ at $x$ *almost surely*, for *all* inputs $x$, when $f$ satisfies some qualification condition. Their work, however, could not provide the correctness of the current autodiff systems, because those systems do not use their algorithm. Bolte and Pauwels [6] presented a result closest to ours: standard autodiff algorithms correctly compute the *standard derivative* of $f$ at $x$ *always*, for *almost all* inputs $x$, if $f$ is definable [43]. To do so, they extended Clarke subdifferential to define a new notion of derivatives, called conservative fields, and showed that autodiff algorithms compute conservative fields and the fields coincide with the standard derivatives almost everywhere. We point out three key differences between these works and ours. First, they use highly advanced mathematical tools (e.g., variational analysis [36]). Thanks to those powerful tools, they could prove more results than ours, such as the almost-sure convergence of the stochastic gradient descent with conservative fields [6], but they are less accessible than our work, which requires much simpler mathematics (probably at an undergraduate level) but still gives the correctness proof of autodiff systems. Second, subdifferentials and conservative fields are difficult to be generalised to higher-order derivatives, because they are defined in terms of a *single set-valued* function. In contrast, our intensional derivatives are defined in terms of a *set* of *standard* (i.e., singleton-valued) functions, and thus have a natural generalisation to higher-order derivatives. Lastly, different function classes

are considered: those works apply to locally Lipschitz continuous functions with additional properties (e.g., definability), whereas ours applies to PAP functions. We point out that PAP functions include many non-locally-Lipschitz functions, such as discontinuous $x \in \mathbb{R} \longmapsto \mathbb{1}[x > 0]$ and continuous $x \in \mathbb{R} \longmapsto \sqrt{|x|}$; the opposite direction is also true (e.g., the $\lambda$-Cantor functions for $\lambda \in (0, 1)$).

The PAP property is similar to some other piecewise properties, such as piecewise smoothness [47] which inspired PAP, and piecewise analyticity [20]. The PAP, however, differs from the others in a few ways: unlike piecewise smoothness, it allows piecewise functions to be defined on open subsets of $\mathbb{R}^n$ (not just on $\mathbb{R}^n$), but restricts them to be analytic (rather than smooth); unlike piecewise analyticity, it allows piecewise domains to be countably many (not just finitely many), but restricts them to be analytic (rather than unrestricted). Unlike prior works, we also give an in-depth study of the PAP property, such as its closedness under operations (e.g., composition) and its relationship to other properties (e.g., almost-everywhere differentiability). The property of almost-everywhere differentiability has been studied as well for programs and functions (that need not be related to programs). E.g., it is shown to hold for all programs in some programming languages [20, 30], and for all piecewise smooth functions [47]. We prove the property for all PAP functions and utilise it in proving that higher-order intensional derivatives agree with standard derivatives almost everywhere.

Recently and concurrently with this work, Bolte and Pauwels [7] studied some concepts and results similar to ours. They proposed a new class of functions and a new notion of derivatives, called elementary selections and selection derivatives, which roughly correspond to our PAP functions and intensional derivatives; and proved properties of those new functions and derivatives, which roughly correspond to our Propositions 8 to 11 and Theorem 12. Although having some similarities, our work and their work have three key differences, complementing each other. First, their work is applicable to a strictly smaller class of functions than ours, as any elementary selection is PAP (and locally Lipschitz) but not vice versa. Second, it considers selection derivatives of first order only, whereas our work considers intensional derivatives of higher orders as well. Third, their work provides some results not in our work (e.g., convergence of stochastic gradient descent with selection derivatives), and vice versa (e.g., the results in §2).

We conclude the section by discussing a few remarks on a notion of correctness for autodiff systems. First, there can be multiple different correctness conditions of autodiff systems and we do not claim that our correctness condition (Definition 1 or Corollary 13) is "the" gold standard. Rather we are just suggesting "a" correctness condition that can serve as a reasonable (possibly minimal) requirement for existing and future autodiff systems. Second, as important as the correctness of autodiff systems is the correctness of the applications built upon autodiff systems (e.g., gradient descent, variational inference, and Hamiltonian Monte Carlo), but the latter might not necessarily follow from the former. For example, consider an autodiff system that is correct in the sense of Corollary 13 (i.e., computes an intensional derivative), and a gradient descent algorithm that follows the "derivative", computed by the system, of an objective function $f : \mathbb{R} \to \mathbb{R}$ to minimise $f$. Suppose that $f$ is given by $f(x) = x^2$ if $x \neq 1$, and 1 if $x = 1$; the autodiff system computes $df(x) = 2x$ if $x \neq 1$, and 0 if $x = 1$, as the "derivative" of $f$; and the gradient descent starts at $x = 1$.[1] Then, the gradient descent gets stuck at $x = 1$ forever since $df(1) = 0$, even though 1 is never a critical point of $f$. This example illustrates that the correctness of autodiff systems does not necessarily imply the correctness of gradient descents built upon those systems (where the latter correctness is given by that gradient descents should converge to Clarke critical points).[2] Nevertheless, the two notions of correctness discussed so far, one for autodiff systems and the other for applications using autodiff systems, address two separate issues and this paper is mainly about the former notion of correctness.

## Broader Impact

This work focuses mainly on theoretical aspects of autodiff systems. In particular, we formally prove that the systems, though developed to handle differentiable functions, remain correct even when applied to non-differentiable functions. Our result justifies, at least in part, the current situation in machine learning, in which the systems are frequently applied to non-differentiable functions without

much consideration to their correctness under such out-of-scope use cases. Other than the justification, this work does not present any other foreseeable societal consequence due to its theoretical nature.

## Acknowledgments and Disclosure of Funding

We thank anonymous reviewers for their insightful and constructive comments. Lee, Yang, and Yu were supported by the Engineering Research Center Program through the National Research Foundation of Korea (NRF) funded by the Korean Government MSIT (NRF-2018R1A5A1059921), and also by Next-Generation Information Computing Development Program through the National Research Foundation of Korea (NRF) funded by the Ministry of Science, ICT (2017M3C4A7068177). Rival was supported by a Facebook gift and by the European Research Council (ERC) under the European Union's Horizon 2020 research and innovation programme (grant agreement No 825492).

## Footnotes

[1]We thank an anonymous reviewer for coming up with this example.

[2]On the other hand, we conjecture that for any PAP and locally Lipschitz $f : \mathbb{R}^n \to \mathbb{R}$, if a gradient descent follows an intensional derivative of $f$ and starts at *randomly chosen* initial points, then it would be correct *almost surely*. A partial result confirming the conjecture was proven recently in [7], concurrently with this work.

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
