[Supplementary Material]

# Supplementary Material:
# On Correctness of Automatic Differentiation
# for Non-Differentiable Functions

## A   Comments on Results in §2

### A.1   Comments on the proof of Proposition 1

First, we elaborate on our proof in Proposition 1 that $g$ is continuous on $(0,1)$. Since $g$ is continuous on $(0,1) \setminus C_1$ by its construction, we only need to show that $g$ is continuous on $C_1$. Consider any $x \in C_1$ and $\epsilon > 0$. It suffices to show that there is $\delta \in (0,x)$ such that

$$0 < |x - x'| < \delta \quad \Longrightarrow \quad |g(x) - g(x')| = |g(x')| = g(x') < \epsilon. \tag{1}$$

Let $k > 0$ be an integer with $2^{-k} < \epsilon$. Consider the set

$$S = \{x' \in (0,1) \setminus C_1 \mid g(x') \geq 2^{-k}\}.$$

By the construction of $g$, $S$ is the union of some finitely many closed intervals in $(0,1)$ that do not contain $x$. (Note that each of those closed intervals is contained in an open interval removed at some $k'(\leq k)$-th step of $g$'s construction.) Hence,

$$\delta = \min \left( \inf_{x' \in S} |x - x'|, \ x \right)$$

is positive. We now show that $\delta$ satisfies (1). Consider any $x'$ with $0 < |x - x'| < \delta$. If $x' \in C_1$, then $g(x') = 0$. If $x' \notin C_1$, then $x' \in (0,1) \setminus C_1$ and $x' \notin S$ by the definition of $\delta$, and thus $g(x') < 2^{-k} < \epsilon$ by the definition of $S$ and $k$. Hence, (1) holds and this completes the proof.  □

Second, we elaborate on our proof in Proposition 1 that $g \circ f$ is not differentiable on $C_{1/2}$. Consider any $x \in C_{1/2}$. It suffices to show that for any $\delta \in (0,x)$, there exist $x_1, x_2 \in (x - \delta, x + \delta) \setminus \{x\}$ such that

$$\left| \frac{(g \circ f)(x) - (g \circ f)(x_1)}{x - x_1} \right| = 0 \quad \text{and} \quad \left| \frac{(g \circ f)(x) - (g \circ f)(x_2)}{x - x_2} \right| > 1. \tag{2}$$

Consider any $\delta \in (0,x)$. Since $x \in C_{1/2}$ is a limit point of $C_{1/2}$, there exists $x_1 \in (x-\delta, x+\delta) \setminus \{x\}$ with $x_1 \in C_{1/2}$. For this $x_1$, the first equality in (2) holds, since $(g \circ f)(C_{1/2}) = g(C_1) = \{0\}$. To find $x_2$, let $k > 0$ be an integer such that

$$d(k) = \frac{1}{2} \cdot \frac{1}{2^k} + \frac{3}{4} \cdot \frac{1}{3^k} < \delta \quad \text{and} \quad \frac{3}{4} \cdot \left(\frac{2}{3}\right)^k < \frac{1}{2}. \tag{3}$$

We claim that there exists $x_2 \in (0,1)$ such that

$$0 < |x - x_2| \leq d(k) \quad \text{and} \quad (g \circ f)(x_2) = 2^{-k}. \tag{4}$$

If the claim holds, then

$$\left| \frac{(g \circ f)(x) - (g \circ f)(x_2)}{x - x_2} \right| \geq 2^{-k} / d(k) \qquad \text{(by (4) and } (g \circ f)(x) = 0)$$

$$= 2^{-k} \Big/ \left( \frac{1}{2} \cdot \frac{1}{2^k} + \frac{3}{4} \cdot \frac{1}{3^k} \right) \qquad \text{(by the equality in (3))}$$

$$= 1 \Big/ \left( \frac{1}{2} + \frac{3}{4} \cdot \left(\frac{2}{3}\right)^k \right)$$

$$> 1 \Big/ \left( \frac{1}{2} + \frac{1}{2} \right) = 1 \qquad \text{(by the inequality in (3)),}$$

and thus the second equality in (2) holds. Hence, finding $x_2 \in (0, 1)$ satisfying (4) completes the proof. We now show that such $x_2$ exists. Consider the situation right after the $k$-th step of $C_{1/2}$'s construction is performed. Then, the total length of the closed intervals that still remain is

$$1 - \frac{1}{2} \cdot \left( \frac{1}{3^1} \cdot 2^0 + \frac{1}{3^2} \cdot 2^1 + \cdots + \frac{1}{3^k} \cdot 2^{k-1} \right) = \frac{1}{2} \cdot \left( 1 + \left( \frac{2}{3} \right)^k \right),$$

so the length of each of those closed intervals is $\frac{1}{2}(2^{-k} + 3^{-k})$, since those closed intervals have the same length and there are $2^k$ such intervals. Due to this, and by the construction of $C_{1/2}$, there is some open interval $I$ that is removed exactly at the $k$-th step of $C_{1/2}$'s construction and satisfies

$$\mathrm{dist}(x, I) \leq \frac{1}{2} \cdot \left( \frac{1}{2^k} + \frac{1}{3^k} \right).$$

Let $x_2 \in (0, 1)$ be the midpoint of $I$. By the construction of $f$ and $g$, we have $(g \circ f)(x_2) = 2^{-k}$. Furthermore, since the length of $I$ is $3^{-k}/2$, we have

$$|x - x_2| \leq \mathrm{dist}(x, I) + \frac{1}{2} \cdot \mathrm{len}(I) \leq \frac{1}{2} \cdot \left( \frac{1}{2^k} + \frac{1}{3^k} \right) + \frac{1}{4} \cdot \frac{1}{3^k} = d(k).$$

Hence, $x_2 \in (0, 1)$ satisfies (4), and this concludes the proof. $\qquad\square$

Next, we make a remark on non-differentiable inputs of $f$, $g$, and $g \circ f$ in the proof. One might guess that $f$ should be non-differentiable exactly on $C_{1/2}$, given that $f$ maps $(0, 1) \setminus C_{1/2}$ onto $(0, 1) \setminus C_1$ in a linear way and maps $C_{1/2}$ onto $C_1$ in a non-smooth-looking way. Surprisingly, the guess is wrong: $f$ is in fact non-differentiable only on a measure-zero subset of $C_{1/2}$. On the other hand, $g$ and $g \circ f$ are non-differentiable exactly on $C_1$ and $C_{1/2}$, respectively. The proof that $g$ is non-differentiable on $C_1$ is similar to the above proof that $g \circ f$ is non-differentiable on $C_{1/2}$, and thus we omit it.

Finally, we connect the examples in the proof with our results in §3. Both $f$ and $g$ are shown to be non-PAP (Proposition 5). Hence, our results do not guarantee that $g \circ f$ is PAP and so almost-everywhere differentiable. In fact, $g \circ f$ is non-PAP, since $g \circ f$ is not almost-everywhere differentiable.

### A.2 Comments on the proof of Proposition 2

We explain how the counterexample in the proof does not contradict to our results in §3. The functions $f$ and $g$ are PAP (and thus $g \circ f$ is so). Although $g'$ is undefined at 0, we can extend it to an intensional derivative $dg \in \partial_\bullet g$ such that $dg$ is defined everywhere (even at 0) and coincides with $g'$ at all but countably many inputs. With such $dg$, the following version of the chain rule holds almost everywhere:

$$(g \circ f)'(x) = dg(f(x)) \cdot f'(x) \quad \text{for almost all } x \in (0, 1).$$

This is because we have the chain rule for intensional derivatives and and these intensional derivatives and standard derivatives coincide almost everywhere (Propositions 8 and 10).

### A.3 Comments on the proof of Proposition 3

The functions $f$, $g$, and $g \circ f$ in the proof do not contradict to our results. Neither $f$ nor $g$ is a PAP function (Proposition 5). Hence, our results do not guarantee the validity of our version of the chain rule for $g \circ f$.

## B Comments on and Proofs for Results in §3

Let $\mathcal{X}, \mathcal{X}_f, \mathcal{X}_g \subseteq \mathbb{R}^n$ and $\mathcal{Y} \subseteq \mathbb{R}^m$ be arbitrary sets.

### B.1 Comments on the proof of Proposition 5

We prove the following argument used in the proof of Proposition 5: the functions listed in the proof satisfy the sufficient condition (i) or (ii) mentioned in the proof.

First, consider (i). To show that a function $h : \mathcal{X} \to \mathbb{R}^m$ satisfies (i) with $k = 2$, it suffices to show the claim that the set

$$S = \{x \in \mathcal{X} \mid h' \text{ is undefined or discontinuous at } x\}$$

has positive measure. For the $\lambda$-Cantor function $\phi_\lambda$ with $\lambda \in (0, 1)$, $S$ is a full measure subset of $C_\lambda$ due to the following: $\phi'_\lambda(x) = 1/(1 - \lambda) \neq 0$ for almost all $x \in C_\lambda$; $\phi'_\lambda(x) = 0$ for all $x \notin C_\lambda$; and $C_\lambda \subset (0, 1)$ has no interior. Since $C_\lambda$ has measure $1 - \lambda > 0$, the claim holds for $\phi_\lambda$. For $f$ in the proof of Proposition 1, $S$ is a full measure subset of $C_{1/2}$ due to similar reasons. So the claim holds for $f$. For Volterra's function, $S$ is known to have positive measure [18, Example 8.35]. So the claim holds for Volterra's function.

Next, consider (ii). Observe that the 1-Cantor function and $g$ in the proof of Proposition 1 are both defined on $(0, 1) \subseteq \mathbb{R}$ and non-differentiable exactly on $C_1$ (for the non-differentiability of $g$, see §A.1). Since $C_1$ is uncountable, the two functions satisfy (ii) with $k = 1$. This completes the proof of the argument. $\qquad\square$

As a side note, we remark that Volterra's function $V$ is more pathological than the $\lambda$-Cantor function $\phi_\lambda$ for $\lambda \in (0, 1)$, in that $V'$ is discontinuous on a set of positive measure even though $V$ is differentiable everywhere and $V'$ is bounded. Contrast this with the fact that $\phi'_\lambda$ is also discontinuous on a set of positive measure, but $\phi_\lambda$ is differentiable just almost everywhere, not everywhere. In fact, there even exists a more pathological function $W : (0, 1) \to \mathbb{R}$ such that $W$ is differentiable everywhere and $W'$ is bounded, but $W'$ is discontinuous almost everywhere [10, Exercise 5:5.5]. Certainly, $W$ is an another example for Proposition 5: it is continuous and differentiable almost everywhere, but not PAP.

## B.2 Interior and subinterior of analytic partition

**Definition 9** (Interior of Analytic Partition). *Let $A = \{A^i\}_{i \in [I]}$ be an analytic partition of $\mathcal{X}$. The **interior** of A, $\mathrm{int}(A)$, is defined by the following open subset of $\mathbb{R}^n$:*

$$\mathrm{int}(A) = \bigcup_{i \in [I]} \mathrm{int}(A^i; \mathbb{R}^n),$$

*where $\mathrm{int}(A^i; \mathbb{R}^n)$ denotes the largest open subset of $\mathbb{R}^n$ that is included in $A^i$.*

**Definition 10** (Subinterior of Analytic Partition). *Let $A = \{A^i\}_{i \in [I]}$ be an analytic partition of $\mathcal{X}$ with $I \in (\mathbb{Z}_{>0} \cup \{\infty\})$. Suppose that a subset $\mathcal{X}' \subseteq \mathcal{X}$ and a partition $B = \{B^t\}_{t \in [T]}$ of $\mathcal{X}'$ satisfy the following conditions:*

(i) *$T \in (\mathbb{Z}_{>0} \cup \{\infty\})$.*
(ii) *For all $t \in [T]$, $B^t$ is **subanalytic**. That is, for all $t \in [T]$, there exist $J_t, L_t \in \mathbb{Z}_{\geq 0}$ and analytic functions $g^+_{t,j} : \mathcal{X}^+_{t,j} \to \mathbb{R}$ and $g^-_{t,l} : \mathcal{X}^-_{t,l} \to \mathbb{R}$ over open domains $\mathcal{X}^+_{t,j}, \mathcal{X}^-_{t,l} \subseteq \mathbb{R}^n$ ($j \in [J_t]$, $l \in [L_t]$) such that: $\mathcal{X}^-_{t,l}$ is connected in $\mathbb{R}^n$ and $(g^-_{t,l})^{-1}(\{0\}) \neq \mathcal{X}^-_{t,l}$ for all $l \in [L_t]$; and*

$$B^t = \Big( \bigcap_{j \in [J_t]} (g^+_{t,j})^{-1}(\mathbb{R}_{>0}) \Big) \cap \Big( \bigcap_{l \in [L_t]} (g^-_{t,l})^{-1}(\mathbb{R}_{<0}) \Big). \tag{5}$$

(iii) *Let $C^t \subseteq \mathbb{R}^n$ be the set defined as follows:*

$$C^t = \Big( \bigcap_{j \in [J_t]} (g^+_{t,j})^{-1}(\mathbb{R}_{>0}) \Big) \cap \Big( \bigcap_{l \in [L_t]} (g^-_{t,l})^{-1}(\mathbb{R}_{\leq 0}) \Big).$$

*Then, $C = \{C^t\}_{t \in [T]}$ is a finer partition of $\mathcal{X}$ than $\{A^i\}_{i \in [I]}$. That is, $C$ is a partition of $\mathcal{X}$, and for all $t \in [T]$, $C^t \subseteq A^i$ for some $i \in [I]$.*

*We call the set $\mathcal{X}'$ a **subinterior** of A, and the partition B a **subanalytic** partition of $\mathcal{X}'$. We use $\mathrm{subint}(A)$ to denote the set of all subinteriors of A.*

**Lemma 14.** *Let $A = \{A^i\}_{i \in [I]}$ be an analytic partition of $\mathcal{X}$ with $I \in (\mathbb{Z}_{>0} \cup \{\infty\})$. Then, $\mathrm{subint}(A) \neq \varnothing$. Furthermore, for any $\mathcal{X}' \in \mathrm{subint}(A)$, the following hold:*

(a) *There is an analytic partition $B = \{B^t\}_{t \in [T]}$ of $\mathcal{X}'$ with $T \in (\mathbb{Z}_{>0} \cup \{\infty\})$ and $\mathrm{int}(B) = \mathcal{X}'$.*

*(b)* $\mathcal{X}' \subseteq \mathrm{int}(A)$ *and* $\mathcal{X}'$ *is open in* $\mathbb{R}^n$.
*(c)* $\mathcal{X} \setminus \mathcal{X}'$ *is contained in some measure-zero set.*
*(d)* $\mathcal{X} \setminus \mathrm{int}(A)$ *is contained in some measure-zero set.*

*Proof.* We first prove that $\mathrm{subint}(A) \neq \varnothing$. Consider any $i \in [I]$. Since $A^i$ is analytic, there exist $J_i, L_i \in \mathbb{Z}_{>0}$ and analytic functions $g_{i,j}^+ : \mathcal{X}_{i,j}^+ \to \mathbb{R}$ and $g_{i,l}^- : \mathcal{X}_{i,l}^- \to \mathbb{R}$ over open domains $\mathcal{X}_{i,j}^+, \mathcal{X}_{i,l}^- \subseteq \mathbb{R}^n$ $(j \in [J_i], l \in [L_i])$ such that

$$A^i = \Big( \bigcap_{j \in [J_i]} (g_{i,j}^+)^{-1}(\mathbb{R}_{>0}) \Big) \cap \Big( \bigcap_{l \in [L_i]} (g_{i,l}^-)^{-1}(\mathbb{R}_{\leq 0}) \Big).$$

We use the following fact: any open set in $\mathbb{R}^n$ is a union of countably many *open balls* in $\mathbb{R}^n$, thereby a union of countably many *disjoint connected* open sets in $\mathbb{R}^n$. For every $l \in [L_i]$, since $\mathcal{X}_{i,l}^-$ is open in $\mathbb{R}^n$, there exists a partition $\{\mathcal{X}_{i,\langle l,t_l \rangle}^-\}_{t_l \in [\infty]}$ of $\mathcal{X}_{i,l}^-$ such that $\mathcal{X}_{i,\langle l,t_l \rangle}^-$ is connected and open in $\mathbb{R}^n$. For each $\langle t_1, \ldots, t_{L_i} \rangle \in [\infty]^{L_i}$, let

$$B^{i,\langle t_1,\ldots,t_{L_i} \rangle} = \Big( \bigcap_{j \in [J_i]} (g_{i,j}^+)^{-1}(\mathbb{R}_{>0}) \Big) \cap \Big( \bigcap_{l \in [L_i]} (g_{i,\langle l,t_l \rangle}^-)^{-1}(\mathbb{R}_{<0}) \Big)$$

$$C^{i,\langle t_1,\ldots,t_{L_i} \rangle} = \Big( \bigcap_{j \in [J_i]} (g_{i,j}^+)^{-1}(\mathbb{R}_{>0}) \Big) \cap \Big( \bigcap_{l \in [L_i]} (g_{i,\langle l,t_l \rangle}^-)^{-1}(\mathbb{R}_{\leq 0}) \Big)$$

where $g_{i,\langle l,t_l \rangle}^- : \mathcal{X}_{i,\langle l,t_l \rangle}^- \to \mathbb{R}$ denotes the restriction of $g_{i,l}^-$ to $\mathcal{X}_{i,\langle l,t_l \rangle}^-$. Here, if $(g_{i,\langle l,t_l \rangle}^-)^{-1}(\{0\}) = \mathcal{X}_{i,\langle l,t_l \rangle}^-$ for some $i, l, t_l$, then we set $g_{i,\langle l,t_l \rangle}^-$ to the constant function $-1$ on the domain $\mathcal{X}_{i,\langle l,t_l \rangle}^-$. Then, every $g_{i,\langle l,t_l \rangle}^-$ is analytic on its connected open domain $\mathcal{X}_{i,\langle l,t_l \rangle}^-$, and $(g_{i,\langle l,t_l \rangle}^-)^{-1}(\{0\}) \neq \mathcal{X}_{i,\langle l,t_l \rangle}^-$. Finally, let

$$\mathcal{X}' = \bigcup_{i \in [I], \, \langle t_1,\ldots,t_{L_i} \rangle \in [\infty]^{L_i}} B^{i,\langle t_1,\ldots,t_{L_i} \rangle}$$

$$B = \{ B^{i,\langle t_1,\ldots,t_{L_i} \rangle} \mid i \in [I], \, \langle t_1, \ldots, t_{L_i} \rangle \in [\infty]^{L_i} \}$$

$$C = \{ C^{i,\langle t_1,\ldots,t_{L_i} \rangle} \mid i \in [I], \, \langle t_1, \ldots, t_{L_i} \rangle \in [\infty]^{L_i} \}.$$

Then, $\mathcal{X}'$ is a subinterior of $A$, and $B$ is a subanalytic partition of $\mathcal{X}'$, because of the following:

- $B$ is a partition of $\mathcal{X}'$.
- $\{ \langle i, t_1, \ldots, t_{L_i} \rangle \mid i \in [I], \, \langle t_1, \ldots, t_{L_i} \rangle \in [\infty]^{L_i} \}$ is a countable set.
- For all $i \in [I]$ and $\langle t_1, \ldots, t_{L_i} \rangle \in [\infty]^{L_i}$, $B^{i,\langle t_1,\ldots,t_{L_i} \rangle}$ is subanalytic.
- $C$ is a finer partition of $\mathcal{X}$ than $A$. This holds because $\{A^i\}_{i \in [I]}$ is a partition of $\mathcal{X}$, and $\{C^{i,\langle t_1,\ldots,t_{L_i} \rangle}\}_{\langle t_1,\ldots,t_{L_i} \rangle \in [\infty]^{L_i}}$ is a partition of $A^i$ for all $i \in [I]$, by its construction.

This completes the proof that $\mathrm{subint}(A) \neq \varnothing$.

We now prove the remaining claims. Let $\mathcal{X}' \in \mathrm{subint}(A)$ and $B = \{B^t\}_{t \in [T]}$ be a subanalytic partition of $\mathcal{X}'$ that satisfies the equations in Definition 10.

Proof of (a). By the definition of subanalytic partition, and since $h(x) < 0 \iff -h(x) > 0$ for any function $h$ and input $x$, $B$ is an analytic partition of $\mathcal{X}'$ with $T \in (\mathbb{Z}_{>0} \cup \{\infty\})$. We argue that for any $t \in [T]$, $B^t$ is open in $\mathbb{R}^n$. Recall the equation (5):

$$B^t = \Big( \bigcap_{j \in [J_t]} (g_{t,j}^+)^{-1}(\mathbb{R}_{>0}) \Big) \cap \Big( \bigcap_{l \in [L_t]} (g_{t,l}^-)^{-1}(\mathbb{R}_{<0}) \Big).$$

Since $g_{t,j}^+ : \mathcal{X}_{t,j}^+ \to \mathbb{R}$ and $g_{t,l}^- : \mathcal{X}_{t,l}^- \to \mathbb{R}$ are continuous, and $\mathbb{R}_{>0}$ and $\mathbb{R}_{<0}$ are open in $\mathbb{R}$, we have that $(g_{t,j}^+)^{-1}(\mathbb{R}_{>0})$ and $(g_{t,l}^-)^{-1}(\mathbb{R}_{<0})$ are open in $\mathcal{X}_{t,j}^+$ and $\mathcal{X}_{t,l}^-$, respectively, by the definition of continuity. Furthermore, since $\mathcal{X}_{t,j}^+$ and $\mathcal{X}_{t,l}^-$ are open in $\mathbb{R}^n$, we have that $(g_{t,j}^+)^{-1}(\mathbb{R}_{>0})$ and $(g_{t,l}^-)^{-1}(\mathbb{R}_{<0})$ are open in $\mathbb{R}^n$ as well. Since any finite intersection of open subsets is again open, $B^t$ is open in $\mathbb{R}^n$. Hence,

$$\mathrm{int}(B) = \bigcup_{t \in [T]} \mathrm{int}(B^t; \mathbb{R}^n) = \bigcup_{t \in [T]} B^t = \mathcal{X}'.$$

Proof of (b). We continue the proof from (a). Since $B^t$ is open in $\mathbb{R}^n$, and $B^t \subseteq A^i$ for some $i \in [I]$ (by the definition of subanalytic partition), we have $B^t \subseteq \text{int}(A^i; \mathbb{R}^n)$. From this, we obtain

$$\mathcal{X}' = \bigcup_{t \in [T]} B^t \subseteq \bigcup_{i \in [I]} \text{int}(A^i; \mathbb{R}^n) = \text{int}(A).$$

Moreover, since any union of open sets is again open, $\mathcal{X}' = \bigcup_{t \in [T]} B^t$ is open in $\mathbb{R}^n$.

Proof of (c). For this, we use the following theorem [32]: for any open connected $U \subseteq \mathbb{R}^n$ and analytic function $f : U \to \mathbb{R}$, the zero set $\{x \in U \mid f(x) = 0\}$ of $f$ is either $U$ or contained in some measure-zero set. Observe that

$$
\begin{aligned}
\mathcal{X} \setminus \mathcal{X}' &= \Big( \bigcup_{t \in [T]} C^t \Big) \setminus \Big( \bigcup_{t \in [T]} B^t \Big) \\
&= \bigcup_{t \in [T]} \Big( C^t \setminus B^t \Big) \\
&\subseteq \bigcup_{t \in [T]} \Big( \bigcap_{l \in [L_t]} (g_{t,l}^-)^{-1}(\mathbb{R}_{\leq 0}) \setminus \bigcap_{l \in [L_t]} (g_{t,l}^-)^{-1}(\mathbb{R}_{<0}) \Big) \\
&\subseteq \bigcup_{t \in [T]} \bigcup_{l \in [L_t]} \Big( (g_{t,l}^-)^{-1}(\mathbb{R}_{\leq 0}) \setminus (g_{t,l}^-)^{-1}(\mathbb{R}_{<0}) \Big) \\
&= \bigcup_{t \in [T]} \bigcup_{l \in [L_t]} (g_{t,l}^-)^{-1}(\{0\}).
\end{aligned}
$$

Since each $g_{t,l}^-$ is analytic, and not everywhere-zero, on its connected open domain $\mathcal{X}_{t,l}^-$ (by the definition of subanalytic partition), the above theorem and equation imply that $(g_{t,l}^-)^{-1}(\{0\})$ is contained in some measure-zero set. Since any countable union of measure-zero sets has measure zero, $\mathcal{X} \setminus \mathcal{X}'$ is contained in some measure-zero set.

Proof of (d). This follows immediately from (b) and (c). $\qquad \square$

### B.3 Proofs of Proposition 4 and Proposition 8 (part I)

We remind the reader that the notation $D(f)(x)$ means the standard derivative of $f$ at $x$.

**Definition 11** (Interior of PAP Representation). *Let $\gamma = \{\langle A^i, f^i \rangle\}_{i \in [I]}$ be a PAP representation from $\mathcal{X}$ to $\mathcal{Y}$. The **interior** and **subinterior** of $\gamma$ are defined by:*

$$\text{int}(\gamma) = \text{int}(\{A^i\}_{i \in [I]}), \qquad \text{subint}(\gamma) = \text{subint}(\{A^i\}_{i \in [I]}).$$

**Lemma 15.** *Let $f : \mathcal{X} \to \mathcal{Y}$ be a PAP function, $\gamma$ be a PAP representation of $f$, and $k \in \mathbb{Z}_{\geq 0}$. Then, for all $x \in \text{int}(\gamma)$, $f$ has the $k$-th order standard derivative at $x$. Furthermore, the derivative agrees with the $k$-th order intensional derivative of $\gamma$ at $x$:*

$$D^{(k)}(f)(x) = \langle\!\langle D^{(k)}(\gamma) \rangle\!\rangle(x) \quad \text{for all } x \in \text{int}(\gamma),$$

*where $F^{(k)}$ denotes the $k$-time composition of the operator $F$.*

*Proof.* Consider any $x \in \text{int}(\gamma)$. By the definition of $\text{int}(\gamma)$, we have $x \in \text{int}(A^i; \mathbb{R}^n)$ for some $i \in [I]$. So there exists an open neighbourhood $U \subseteq \mathbb{R}^n$ of $x$ such that $U \subseteq A^i$. Since $\gamma$ is a representation of $f$ and $U \subseteq A^i$, we have $f = f^i$ on $U$ for the $i$-th component function of $\gamma$. Now focus on $f^i$. Since $f^i$ is analytic on $U$ (due to $\gamma$ being PAP), $f^i$ is infinitely differentiable on $U$ and, in particular, has the $k$-th order standard derivative at $x$, namely $D^{(k)}(f^i)(x)$. By the definition of intensional derivative, $D^{(k)}(f^i)(x) = \langle\!\langle D^{(k)}(\gamma) \rangle\!\rangle(x)$. Since $x \in U$, $U$ is open in $\mathbb{R}^n$, and $f = f^i$ on $U$, we obtain that $D^{(k)}(f)(x) = D^{(k)}(f^i)(x) = \langle\!\langle D^{(k)}(\gamma) \rangle\!\rangle(x)$. $\qquad \square$

**Proposition 16.** *Let $f : \mathcal{X} \to \mathcal{Y}$ be a PAP function and $k \in \mathbb{Z}_{\geq 0}$. Then, $f$ has the $k$-th order standard derivative almost everywhere. Moreover, the first-order standard derivative $Df$ agrees with any first-order intensional derivative $df \in \partial_\bullet f$ almost everywhere.*

*Proof.* By Lemma 14(d), the interior of any PAP representation of $f$ has the full measure in $\mathcal{X}$. Hence, Lemma 15 implies the first claim. For the second claim, let $df \in \partial_\bullet f$. By the definition of $\partial_\bullet f$, there exists a PAP representation $\gamma$ of $f$ such that $df = \langle\!\langle D\gamma \rangle\!\rangle$. Applying Lemma 15 to $f, \gamma$, and $k = 1$ gives the second claim, since $\text{int}(\gamma)$ has the full measure in $\mathcal{X}$. $\qquad \square$

## B.4 Proof of Proposition 8 (part II)

**Lemma 17.** *Let $f : \mathcal{X}_f \to \mathcal{Y}$ and $g : \mathcal{X}_g \to \mathcal{Y}$ be PAP functions, and $\gamma_f$ and $\gamma_g$ be their PAP representations. If $f(x) = g(x)$ for all $x \in \mathrm{int}(\gamma_f) \cap \mathrm{int}(\gamma_g)$, then*

$$D(f)(x) = D(g)(x) \quad \text{for all } x \in \mathrm{int}(\gamma_f) \cap \mathrm{int}(\gamma_g),$$

*where both sides are well-defined for each $x$.*

*Proof.* Let $U = \mathrm{int}(\gamma_f) \cap \mathrm{int}(\gamma_g)$, and consider any $x \in U$. Since $x \in U$ and $U$ is open in $\mathbb{R}^n$, we have $D(f)(x) = D(g)(x)$ if both sides are well-defined. Indeed, they are well-defined by Lemma 15 with $k = 1$, since $\gamma_f$ and $\gamma_g$ are PAP representations of $f$ and $g$, respectively. $\qquad\square$

**Lemma 18.** *Let $f : \mathcal{X} \to \mathcal{Y}$ be a PAP function. Then, for any intensional derivative $df \in \partial_\bullet f$, there exists a PAP representation $\gamma_{df}$ of $df$ such that*

$$df(x) = D(f)(x) \quad \text{for all } x \in \mathrm{int}(\gamma_{df}).$$

*Proof.* By the definition of $\partial_\bullet f$, there exists a PAP representation $\gamma_f$ of $f$ such that $df = \langle\!\langle D\gamma_f \rangle\!\rangle$. By Lemma 15 with $k = 1$, we have $D(f)(x) = \langle\!\langle D\gamma_f \rangle\!\rangle(x) = df(x)$ for all $x \in \mathrm{int}(\gamma_f)$. Let $\gamma_{df} = D\gamma_f$. Since $df = \langle\!\langle D\gamma_f \rangle\!\rangle = \langle\!\langle \gamma_{df} \rangle\!\rangle$ and $\gamma_f$ is PAP, $\gamma_{df}$ is a PAP representation of $df$. Moreover, by the definition of $D\gamma_f$, we have $\mathrm{int}(\gamma_{df}) = \mathrm{int}(\gamma_f)$. Hence, the claim holds with $\gamma_{df}$. $\qquad\square$

**Definition 12** (Refinement of Representation). *Let $\gamma_f = \{\langle A^i, f^i \rangle\}_{i \in [I]}$ be a representation of a function from $\mathcal{X}_f$ to $\mathcal{Y}$, and $B = \{B^j\}_{j \in [J]}$ be a partition of $\mathcal{X}_g$. The **refinement** of $\gamma_f$ with $B$ is defined by:*

$$\mathrm{refine}(\gamma_f; B) = \{\langle A^i \cap B^j, f^i \rangle\}_{\langle i,j \rangle \in [I] \times [J]}.$$

*Moreover, for any representation $\gamma_g = \{\langle C^l, g^l \rangle\}_{l \in [L]}$ of a function from $\mathcal{X}_g$ to $\mathcal{Z}$, the **refinement** of $\gamma_f$ with $\gamma_g$ is defined by:*

$$\mathrm{refine}(\gamma_f; \gamma_g) = \mathrm{refine}(\gamma_f; \{C^l\}_{l \in [L]}).$$

**Lemma 19.** *Let $f : \mathcal{X}_f \to \mathcal{Y}$ be a PAP function, $\gamma$ be a PAP representation of $f$, and $B = \{B^j\}_{j \in [J]}$ be an analytic partition of $\mathcal{X}_g$ with $J \in (\mathbb{Z}_{>0} \cup \{\infty\})$. Let $\gamma' = \mathrm{refine}(\gamma; B)$. Then, $\gamma'$ is a PAP representation of $f|_{\mathcal{X}_f \cap \mathcal{X}_g}$ with*

$$\mathrm{int}(\gamma') = \mathrm{int}(\gamma) \cap \mathrm{int}(B).$$

*Proof.* Let $\gamma = \{\langle A^i, f^i \rangle\}_{i \in [I]}$. Since $\gamma$ is PAP and $B$ is an analytic partition, $\{A^i \cap B^j\}_{\langle i,j \rangle \in [I] \times [J]}$ is an analytic partition. Also, since $[J]$ is countable, $[I] \times [J]$ is also countable. Thus, $\gamma'$ is PAP. Since

$$\bigcup_{\langle i,j \rangle \in [I] \times [J]} (A^i \cap B^j) = \Big( \bigcup_{i \in [I]} A^i \Big) \cap \Big( \bigcup_{j \in [J]} B^j \Big) = \mathcal{X}_f \cap \mathcal{X}_g,$$

$\gamma'$ is a representation of $f|_{\mathcal{X}_f \cap \mathcal{X}_g}$. Finally, we obtain the last claim as follows:

$$
\begin{aligned}
\mathrm{int}(\gamma') &= \bigcup_{\langle i,j \rangle \in [I] \times [J]} \mathrm{int}(A^i \cap B^j; \mathbb{R}^n) \\
&= \bigcup_{\langle i,j \rangle \in [I] \times [J]} \mathrm{int}(A^i; \mathbb{R}^n) \cap \mathrm{int}(B^j; \mathbb{R}^n) \\
&= \Big( \bigcup_{i \in [I]} \mathrm{int}(A^i; \mathbb{R}^n) \Big) \cap \Big( \bigcup_{j \in [J]} \mathrm{int}(B^j; \mathbb{R}^n) \Big) \\
&= \mathrm{int}(\gamma) \cap \mathrm{int}(B).
\end{aligned}
$$

For the second equality, we use the following fact: $\mathrm{int}(S_1 \cap S_2; X) = \mathrm{int}(S_1; X) \cap \mathrm{int}(S_2; X)$ for any $S_1, S_2 \subseteq X$. $\qquad\square$

**Lemma 20.** *Let $f : \mathcal{X} \to \mathcal{Y}$ be a PAP function. Consider any PAP representation $\gamma$ of $f$, and any subinterior $\mathcal{X}' \in \mathrm{subint}(\gamma)$. Then, $\mathcal{X}' \subseteq \mathcal{X}$ is open in $\mathbb{R}^n$ and $\mathcal{X} \setminus \mathcal{X}'$ is contained in a measure-zero set. Moreover, for all $k \in \mathbb{Z}_{\geq 0}$, $D^{(k)}(f|_{\mathcal{X}'})$ is a total function on $\mathcal{X}'$, and there exists a PAP representation $\gamma_D^k$ of $D^{(k)}(f|_{\mathcal{X}'})$ such that $\mathrm{int}(\gamma_D^k) = \mathcal{X}'$.*

*Proof.* Let $k \in \mathbb{Z}_{\geq 0}$. By Lemma 14(b) and 14(c), $\mathcal{X}' \subseteq \mathrm{int}(\gamma) \subseteq \mathcal{X}$ is open in $\mathbb{R}^n$ and $\mathcal{X} \setminus \mathcal{X}'$ is contained in a measure-zero set. This proves the first claim. Since $\mathcal{X}' \subseteq \mathrm{int}(\gamma)$, Lemma 15 implies that $D^{(k)}(f)(x)$ exists for all $x \in \mathcal{X}'$. Since $\mathcal{X}'$ is open in $\mathbb{R}^n$, $D^{(k)}(f|_{\mathcal{X}'})(x) = D^{(k)}(f)(x)$ for all $x \in \mathcal{X}'$. This proves the second claim that $D^{(k)}(f|_{\mathcal{X}'})$ is a total function on $\mathcal{X}'$.

We now prove the last claim. By Lemma 14(a), there exists an analytic partition $B = \{B^j\}_{j \in [J]}$ of $\mathcal{X}'$ such that $J \in (\mathbb{Z}_{>0} \cup \{\infty\})$ and $\mathrm{int}(B) = \mathcal{X}'$. Let $\gamma' = \mathrm{refine}(\gamma; B)$. By Lemma 19, $\gamma'$ is a PAP representation of $f|_{\mathcal{X}'}$ with $\mathrm{int}(\gamma') = \mathrm{int}(\gamma) \cap \mathrm{int}(B) = \mathcal{X}'$. Consider
$$\gamma_D^k = D^{(k)}(\gamma').$$
We show that it satisfies the last claim. Since $f|_{\mathcal{X}'} : \mathcal{X}' \to \mathcal{Y}$ is a PAP function with a PAP representation $\gamma'$, Lemma 15 implies that $D^{(k)}(f|_{\mathcal{X}'})(x) = \langle\!\langle D^{(k)}(\gamma')\rangle\!\rangle(x) = \langle\!\langle \gamma_D^k \rangle\!\rangle(x)$ for all $x \in \mathrm{int}(\gamma') = \mathcal{X}'$. Hence, $\gamma_D^k$ is a representation of $D^{(k)}(f|_{\mathcal{X}'})$. The rest of the claim also holds as follows: $\gamma_D^k$ is PAP since $\gamma'$ is PAP; and $\mathrm{int}(\gamma_D^k) = \mathrm{int}(\gamma') = \mathcal{X}'$ by the definition of $D(\gamma')$. $\qquad \square$

**Lemma 21.** *Let $f : \mathcal{X} \to \mathcal{Y}$ be a PAP function and $\mathcal{X}' \subseteq \mathcal{X}$ be a set described in Lemma 20, which has full measure in $\mathcal{X}$. Consider any $k \in \mathbb{Z}_{\geq 0}$. Then, for any $df^k \in \partial_\bullet^k f$, there exists a PAP representation $\gamma_d^k$ of $df^k$ such that*
$$df^k(x) = D^{(k)}(f)(x) \quad \text{for all } x \in \mathrm{int}(\gamma_d^k) \cap \mathcal{X}'.$$

*Proof.* The proof proceeds by induction on $k$. For $k = 0$, we have $df^k = D^{(k)}(f) = f$. So any PAP representation $\gamma_d^k$ of $df^k = f$ satisfies the claim. Now suppose $k > 0$. By the definition of $\partial_\bullet^k f$, there exists $df^{k-1} \in \partial_\bullet^{k-1} f$ such that $df^k \in \partial_\bullet(df^{k-1})$. We construct the desired PAP representation $\gamma_d^k$ as follows. First, focus on $df^{k-1} \in \partial_\bullet^{k-1} f$. By the induction hypothesis on $k-1$ for $df^{k-1}$, there exists a PAP representation $\gamma_d^{k-1}$ of $df^{k-1}$ such that
$$df^{k-1}(x) = D^{(k-1)}(f)(x) \qquad \text{for all } x \in \mathrm{int}(\gamma_d^{k-1}) \cap \mathcal{X}'.$$

By Lemma 20, $D^{(k-1)}(f|_{\mathcal{X}'})$ is a total function on $\mathcal{X}'$ and there exists a PAP representation $\gamma_D^{k-1}$ of $D^{(k-1)}(f|_{\mathcal{X}'})$ such that $\mathrm{int}(\gamma_D^{k-1}) = \mathcal{X}'$. Since $\mathcal{X}'$ is open in $\mathbb{R}^n$ by Lemma 20, $D^{(k-1)}(f) = D^{(k-1)}(f|_{\mathcal{X}'})$ over $\mathcal{X}'$. Combining the two results gives that
$$df^{k-1}(x) = D^{(k-1)}(f|_{\mathcal{X}'})(x) \qquad \text{for all } x \in \mathrm{int}(\gamma_d^{k-1}) \cap \mathrm{int}(\gamma_D^{k-1}),$$

where $\gamma_d^{k-1}$ and $\gamma_D^{k-1}$ are PAP representations of $df^{k-1}$ and $D^{(k-1)}(f|_{\mathcal{X}'})$, respectively. By Lemma 17 applied to this result, we obtain
$$D(df^{k-1})(x) = D^{(k)}(f|_{\mathcal{X}'})(x) \qquad \text{for all } x \in \mathrm{int}(\gamma_d^{k-1}) \cap \mathrm{int}(\gamma_D^{k-1}).$$

Since $D^{(k)}(f|_{\mathcal{X}'}) = D^{(k)}(f)$ over $\mathcal{X}'$ (as $\mathcal{X}'$ is open in $\mathbb{R}^n$), and $\mathrm{int}(\gamma_D^{k-1}) = \mathcal{X}'$, we have
$$D(df^{k-1})(x) = D^{(k)}(f)(x) \qquad \text{for all } x \in \mathrm{int}(\gamma_d^{k-1}) \cap \mathcal{X}'.$$

Next, focus on $df^k \in \partial_\bullet(df^{k-1})$. By Lemma 18 applied to $df^k$, there exists a PAP representation $\gamma_d'^k$ of $df^k$ such that
$$df^k(x) = D(df^{k-1})(x) \qquad \text{for all } x \in \mathrm{int}(\gamma_d'^k).$$
Combining the last two equations, we obtain
$$df^k(x) = D^{(k)}(f)(x) \qquad \text{for all } x \in \mathrm{int}(\gamma_d'^k) \cap \mathrm{int}(\gamma_d^{k-1}) \cap \mathcal{X}'.$$

Now let $\gamma_d^k = \mathrm{refine}(\gamma_d'^k; \gamma_d^{k-1})$. By Lemma 19 applied to $\gamma_d^k$, we have that $\gamma_d^k$ is a PAP representation of $df^k$ with $\mathrm{int}(\gamma_d^k) = \mathrm{int}(\gamma_d'^k) \cap \mathrm{int}(\gamma_d^{k-1})$. From this, we obtain the desired claim:
$$df^k(x) = D^{(k)}(f)(x) \qquad \text{for all } x \in \mathrm{int}(\gamma_d^k) \cap \mathcal{X}'.$$
$\qquad \square$

**Proposition 22.** *Let $f : \mathcal{X} \to \mathcal{Y}$ be a PAP function and $k \in \mathbb{Z}_{\geq 0}$. Then, any $k$-th order intensional derivative $df^k \in \partial_\bullet^k f$ satisfies the following:*
$$df^k(x) = D^{(k)}(f)(x) \quad \text{for almost all } x \in \mathcal{X}.$$

*Proof.* The claim follows from Lemma 21 and the following: $\mathcal{X}'$ and $\mathrm{int}(\gamma_d^k)$ described in Lemma 21 have the full measure in $\mathcal{X}$, by Lemma 21 and Lemma 14(d). $\qquad \square$

### B.5 Additional property on PAP functions

**Proposition 23.** *Let $f : \mathcal{X} \to \mathcal{Y}$ be a PAP function, and $k \in \mathbb{Z}_{>0}$. Let $N_{f,k} \subseteq \mathcal{X}$ be the set defined by $\{x \in \mathcal{X} \mid D^{(k)}(f)(x) \text{ is undefined}\}$. If $n = 1$, then $N_{f,k}$ is countable. But if $n > 1$, then it could be uncountable.*

*Proof.* Suppose $n = 1$. Recall the following two well-known results: (i) if $g : U \to \mathbb{R}$ is an analytic function on an open interval $U \subseteq \mathbb{R}$ and is not everywhere-zero, then its zero set $Z_g = \{x \in U \mid g(x) = 0\}$ contains none of the limit points of $Z_g$ [25, Corollary 1.2.7]; (ii) if $X$ is a second-countable space and $S \subseteq X$ contains none of the limit points of $S$, then $S$ is countable. Since $\mathbb{R}$ is second-countable, every $g$ satisfying the assumption of (i) has at most countably many zeros.

Now return to our claim. By the statement and proof of Lemma 14(c) and Lemma 15, there are countably many analytic functions $\{g_j : U_j \to \mathbb{R}\}_j$ defined over connected open subsets of $\mathbb{R}$, such that every $g_j$ is not everywhere-zero and $N_{f,k} \subseteq \bigcup_j Z_{g_j}$. Since any connected open subset of $\mathbb{R}$ is an open interval, $g_j$ satisfies the assumption of (i), and thus each $Z_{g_j}$ is countable by the above result. Hence, $N_{f,k}$ is countable, since a countable union of countable sets is countable.

Suppose $n > 1$. Consider $f : \mathbb{R}^2 \to \mathbb{R}$ defined by $f(x) = |x_1 - x_2|$, and $k = 1$. Then, $f$ is PAP, since the following is a PAP representation of $f$:

$$\begin{aligned} \{ \quad &\langle \{x \in \mathbb{R}^2 \mid x_1 > x_2\}, x \in \mathbb{R}^2 \longmapsto x_1 - x_2 \rangle, \\ &\langle \{x \in \mathbb{R}^2 \mid x_1 = x_2\}, x \in \mathbb{R}^2 \longmapsto 0 \rangle, \\ &\langle \{x \in \mathbb{R}^2 \mid x_1 < x_2\}, x \in \mathbb{R}^2 \longmapsto x_2 - x_1 \rangle \quad \}. \end{aligned}$$

However, $N_{f,k} = \{\langle x, y \rangle \in \mathbb{R}^2 \mid x = y\}$ is uncountable. $\qquad\square$

## C   Proofs for Results in §4

**Proposition 11.** *For every program $e$, its denotation $[\![e]\!]$ is a PAP function from $\mathbb{R}^N$ to $\mathbb{R}$.*

*Proof.* The proof is by induction on the structure of $e$. The cases of $e \equiv c$ and $e \equiv x_i$ follow from the fact that both constant functions and projections are PAP. For the case of $e \equiv \bar{\mathtt{f}}(e_1, \ldots, e_n)$, we note two facts about PAP functions: the composition of two PAP functions is PAP (Proposition 9); and for PAP functions $g_1, \ldots, g_n : \mathbb{R}^N \to \mathbb{R}$, the function $v \longmapsto \langle g_1(v), \ldots, g_n(v) \rangle$ of type $\mathbb{R}^N \to \mathbb{R}^n$ is PAP again, mainly because any finite intersection of open sets is again open. By these two facts, the claimed property of the proposition holds in this case. The only remaining case is $e \equiv (\mathtt{if} \ (e_1 > 0) \ e_2 \ e_3)$. By induction hypothesis, all of $[\![e_1]\!]$, $[\![e_2]\!]$, and $[\![e_3]\!]$ are PAP. Let $\gamma_1 = \{\langle A^i, f^i \rangle\}_{i \in [I]}$, $\gamma_2 = \{\langle B^j, g^j \rangle\}_{j \in [J]}$, and $\gamma_3 = \{\langle C^k, h^k \rangle\}_{k \in [K]}$ be their PAP representations, and define their conditional composition $\mathrm{cond}(\gamma_1, \gamma_2, \gamma_3)$ as follows:

$$\begin{aligned} \mathrm{cond}(\gamma_1, \gamma_2, \gamma_3) &= \{\langle E^{\langle i,j,k,l \rangle}, t^{\langle i,j,k,l \rangle} \rangle\}_{\langle i,j,k,l \rangle \in ([I] \times [J] \times [K] \times \{0,1\})}, \\ E^{\langle i,j,k,l \rangle} &= A_l^i \cap B^j \cap C^k, \qquad t^{\langle i,j,k,l \rangle} = \mathtt{if} \ (l = 1) \ \mathtt{then} \ g^j \ \mathtt{else} \ h^k \end{aligned}$$

where $A_1^i = \{v \in A^i \mid f^i(v) > 0\}$ and $A_0^i = \{v \in A^i \mid f^i(v) \leq 0\}$. Then, $\{E^{\langle i,j,k,l \rangle}\}_{\langle i,j,k,l \rangle}$ is an analytic partition of $\mathbb{R}^N$, every $t^{\langle i,j,k,l \rangle}$ is an analytic function, and its domain is an open set containing $E^{\langle i,j,k,l \rangle}$. Thus, $\mathrm{cond}(\gamma_1, \gamma_2, \gamma_3)$ is a PAP representation. Furthermore, its evaluation $\langle\!\langle \mathrm{cond}(\gamma_1, \gamma_2, \gamma_3) \rangle\!\rangle$ is equal to $[\![\mathtt{if} \ (e_1 > 0) \ e_2 \ e_3]\!]$. Hence, the proposition holds in this case. $\quad\square$

**Theorem 12.** *If $\widetilde{D}\mathtt{f} \in \partial_\bullet \mathtt{f}$ for all primitive functions $\bar{\mathtt{f}}$, then $[\![e]\!]^\nabla \in \partial_\bullet [\![e]\!]$ for all programs $e$.*

*Proof.* The proof is by induction on the structure of $e$. When $e \equiv c$, the trivial partition $\{\mathbb{R}^N\}$ and the constant function $v \longmapsto c$ form a PAP representation of $[\![e]\!]$. The intensional derivative of this representation is $\{\langle \mathbb{R}^N, v \longmapsto \vec{0}_{1 \times N} \rangle\}$, and its evaluation is $[\![c]\!]^\nabla$, as claimed by the theorem. The other base case is $e \equiv x_i$. We use the trivial partition again with the projection function $v \longmapsto v_i$, and form a PAP representation of $[\![e]\!]$. The intensional derivative of this representation is $\{\langle \mathbb{R}^N, v \longmapsto [\vec{0}_{(i-1) \times 1}; \vec{1}_{1 \times 1}; \vec{0}_{(N-i) \times 1}]^\top \rangle\}$, and its evaluation is $[\![x_i]\!]^\nabla$.

The next case is $e \equiv \bar{\mathtt{f}}(e_1, \ldots, e_n)$. By induction hypothesis, $[\![e_i]\!]^\nabla$ is an intensional derivative of $[\![e_i]\!]$ for every $i \in [n]$. Let $g : \mathbb{R}^N \to \mathbb{R}^n$ and $dg : \mathbb{R}^N \to \mathbb{R}^{n \times N}$ be functions defined by

$g(v) = \langle [\![e_1]\!]v, \ldots, [\![e_n]\!]v \rangle$ and $dg(v) = [[\![e_1]\!]^\nabla v; \ldots; [\![e_n]\!]^\nabla v]$ for all $v$. Then, $dg$ is an intensional derivative of $g$. Also, $[\![\overline{\mathtt{f}}(e_1, \ldots, e_n)]\!] = \mathtt{f} \circ g$. Therefore, by the chain rule for intensional derivative (Proposition 10), the function $v \longmapsto (\widetilde{D}\mathtt{f})(g(v)) \cdot dg(v)$ is an intensional derivative of $[\![\overline{\mathtt{f}}(e_1, \ldots, e_n)]\!] = \mathtt{f} \circ g$. Here we use the assumption that $\widetilde{D}\mathtt{f}$ is an intensional derivative of $\mathtt{f}$. Note that the function is equal to $[\![\overline{\mathtt{f}}(e_1, \ldots, e_n)]\!]^\nabla$. So, the theorem holds.

The last case is $e \equiv (\mathtt{if}\ (e_1 > 0)\ e_2\ e_3)$. By induction hypothesis, $[\![e_i]\!]^\nabla$ is an intensional derivative of $[\![e_i]\!]$ for all $i \in [3]$. Let $\gamma_1, \gamma_2$, and $\gamma_3$ be the PAP representations of $[\![e_1]\!], [\![e_2]\!]$, and $[\![e_3]\!]$ such that

$$\langle\!\langle D\gamma_2 \rangle\!\rangle = [\![e_2]\!]^\nabla \quad \text{and} \quad \langle\!\langle D\gamma_3 \rangle\!\rangle = [\![e_3]\!]^\nabla. \tag{6}$$

Let $\mathrm{cond}(\gamma_1, \gamma_2, \gamma_3)$ be the conditionally composed representation defined in the proof of Proposition 11. Then, it is a PAP representation of $[\![\mathtt{if}\ (e_1 > 0)\ e_2\ e_3]\!]$. But by the equations in (6) and the definitions of $[\![-]\!]^\nabla$, $\mathrm{cond}(-, -, -)$, and $\langle\!\langle D- \rangle\!\rangle$, we have $\langle\!\langle D\mathrm{cond}(\gamma_1, \gamma_2, \gamma_3) \rangle\!\rangle(v) = [\![\mathtt{if}\ (e_1 > 0)\ e_2\ e_3]\!]^\nabla v$ for all $v \in \mathbb{R}^N$. Hence, the theorem holds in this case as well. □

**Corollary 13.** *Assume that an autodiff system for the language in §4 satisfies the two requirements in §4. Then, for each program $e$, there exists an intensional derivative $df$ in $\partial_\bullet [\![e]\!]$ such that if the system performs forward-mode (or reverse-mode) autodiff with a tangent vector $w \in \mathbb{R}^N$ (or a cotangent vector $u \in \mathbb{R}$), it computes the Jacobian-vector product $df(v) \cdot w \in \mathbb{R}$ (or the vector-Jacobian product $u^\top \cdot df(v) \in \mathbb{R}^N$) for every input $v \in \mathbb{R}^N$. Furthermore, the computed entity is the corresponding Jacobian-vector product (or vector-Jacobian product) with the standard derivative of $[\![e]\!]$ for almost all inputs $v \in \mathbb{R}^N$.*

*Proof.* Suppose that the two requirements are met, and let $e$ be a program. Consider the case when the system performs forward-mode autodiff. Let $df = [\![e]\!]^\nabla$. Then, by Theorem 12, $df$ is an intensional derivative in $\partial_\bullet [\![e]\!]$. Moreover, by the second requirement, the output of the system for $e$ with a tangent vector $w \in \mathbb{R}^N$ is $([\![e]\!]^\nabla v) \cdot w = df(v) \cdot w \in \mathbb{R}$ for all inputs $v \in \mathbb{R}^N$. This proves the first part of the corollary. The other part of the corollary follows immediately from Proposition 8. The proof for the case when reverse-mode autodiff is performed is essentially the same, so we omit it. □