[Reviews · NeurIPS 2020]

Review 1

Summary and Contributions: This paper begins with the familiar observation that automatic differentiation systems are often applied to functions that are not actually differentiable (e.g., ReLU), and asks – essentially – why it is that this is OK. In response it contributes: 1. Counter-arguments to the informal but commonly heard answer that “it’s OK because the functions are at least almost everywhere differentiable.” Through a series of counterexamples, the paper shows that (a) programs built by composing a.e.-differentiable functions may not themselves represent a.e.-differentiable functions; (b) even when these compositions are a.e.-differentiable, using the chain rule may require evaluating derivatives that don’t exist; and (c) this can’t be patched up by using arbitrary values whenever an intermediate function is not differentiable. 2. A class of functions (“piecewise analytic under analytic partition”), which includes non-differentiable functions, and a notion of differentiation for this class (“intensional derivatives”), 3. Sufficient conditions for an AD system to be “correct” according to these new definitions (i.e., for the system to compute intensional derivatives of PAP functions built using programming constructs like function calls and if statements). Together, (2) and (3) constitute an answer to “why it’s OK” that AD systems are sometimes applied to non-differentiable functions: their internal logic corresponds to a set of assumptions that are satisfied by PAP functions, and PAP functions are the only functions people write using systems like TensorFlow and PyTorch.

Strengths: Congratulations to the authors on this very enjoyable paper! It is clearly written, easy to digest, and sometimes surprising (I am thinking in particular of the debunking of Claim 3). The proposed solution – PAP and intensional derivatives – is natural and easy to grasp even without much expertise. It is useful to have a name for, and a clear characterization of, a large class of functions for which AD systems can be said to work. More broadly, the notion that functions implemented in programming languages can often be thought of as having nice/desirable properties piecewise, according to a partition that itself has some desirable property, could prove a useful tool for analyzing other features of languages for machine learning (including probabilistic programming languages, for example).

Weaknesses: This paper analyzes the current state-of-the-art in AD systems, with the goal of discovering a sense in which these AD systems can be considered “correct.” As such, its definitions permit and thus overlook the various ways in which today’s AD systems are considered by practitioners to be buggy. In this sense, we might think of this paper as playing the role of “legalese” that implementors of AD systems can hide behind when their tools do unexpected things: 'we are closing this issue,' they can say, 'because the current implementation already returns an intensional derivative.' Knowing how to write that legalese is important, but I wonder if the paper might devote a little more discussion to the ways in which intensional derivatives can be surprising or ‘wrong’ from the user’s perspective. Even when a function is differentiable, it will have intensional derivatives that disagree with its ordinary derivatives at points. It’s true that these occur on a measure-0 set of inputs, but that doesn’t mean users won’t encounter them. This is because the inputs to differentiable loss functions (the initial parameters) may not be drawn randomly from a distribution; they may be set deterministically, e.g., to zero. If I want to optimize f(x) = if x == 0 then 10 else (x – 5)(x – 2), starting at x = 0, I don’t want the AD system to think I am already at an optimum. (This if statement is technically unnecessary, but programs often contain ‘fast paths’ that recognize certain common inputs and then compute the same thing as the ‘slow path’ but more efficiently. Can we formulate a spec for an AD system that permits some forms of non-differentiability but does not get confused by differentiable functions defined in unusual ways?)

Correctness: The claims all look sound to me!

Clarity: I thought the paper was very well-written – it was a pleasure to read.

Relation to Prior Work: I am not very familiar with the relevant related work, but the Related Work section was clearly written and argued persuasively that PAP is a novel contribution, achieving some of the same things that existing work achieves, but with simpler math.

Reproducibility: Yes

Additional Feedback: To the authors: thank you for your clear and thought-provoking responses! I've read them and will still be recommending acceptance.


Review 2

Summary and Contributions: The paper provides theorems why standard autodiff systems still compute correct results when computing derivatives of non-differentiable functions. The paper does this by introducing a new framework for differentiability that is better applicable to standard autodiff tools. It defines the notion of a PAP function (piecewise analytic under analytic partition) and its corresponding derivative (which is coined intensional derivative). Loosely speaking, the core idea is: Cover the domain of a function by countably many regions such that in each region the function is differentiable. Define derivatives of this compound then and use it for the theory. It coincides almost everywhere with the true derivative and in practice, it coincides with what autodiff systems compute. Usually it is argued in standard autodiff systems that non-differentiability is not a problem since it only occurs on a set of measure zero. The paper shows by a number of counterexamples that this reasoning is flawed, especially in the context of function composition. It impressively shows the need for proper, mathematically correct arguments. They are then provided in the paper.

Strengths: The paper addresses some very important and necessary topic. The paper provides and proves important results. The presented theory is sound. Though it will not change the world, mainly due to its theoretic nature, it addresses an important aspect of autodiff systems; their correctness. It is a novel and very contribution. It is highly relevant to the NeurIPS community. As said, it will not change how people will work with autodiff systems but it provides the theoretical grounding that these systems are (usually) correct in the presence of non-differentiable function.

Weaknesses: Nothing.

Correctness: I did not check all theorems fully in-depth. But they seem all correct and sound to me.

Clarity: The paper is very well written. It was a joy to read it! The authors provide proper intuition behind their algorithms and definitions.

Relation to Prior Work: Relation to prior work is properly discussed.

Reproducibility: Yes

Additional Feedback: I like the paper a lot! Very nice theory, very important and necessary result. Very minor issue: It would be nice to have some illustration of the functions used in Proposition 1. The reader will anyway draw an illustration of these functions in order to better understand them. So it would be nice to assist the reader here. The Cantor function might still be present to many readers but the other functions need a bit more thinking. Typo: Line 326: cotangent vector $u \in \R$ -> cotangent vector $u^\top \in \R^N$ ######################## Update after rebuttal ##################### After reading the rebuttal, I am still very fond of this work. As all reviewers agree, it is a very nice and necessary contribution. I will stick to my exceptionally high score.


Review 3

Summary and Contributions: The paper investigates the behavior of autodiff when applied at points where a function is non-differentiable, pointing out that prominent autodiff systems still return a numerical value at such points (as opposed to an error). The paper disproves common misconceptions on the differentiability of composite functions, thereby motivating the need for a more formal framework. To study the behavior of autodiff, the authors introduce the concepts of PAP and intensional derivatives. The first describes primitive functions available to users in autodiff system; the second the object returned by autodiff, as formally proven in the article. Correctness, as defined in the paper, is proven. The article then briefly examines the benefit of treating a function as a PAP, using examples from TensorFlow and PyTorch.

Strengths: The paper raises an important subtlety with regard to autodiff, a ubiquitous tool in machine learning. It is important to understand the behavior of autodiff, as it may not return standard derivatives. I salute the effort to dispel important misconceptions about a.e. differentiability, through the study of three claims and the production of three enlightening counter-examples.

Weaknesses: The authors should explain whether correctness, as defined in the paper, is a satisfying criterion. After all, even if correctness is achieved, is it not problematic that autodiff spits out a solution when the target function is not differentiable? The answer depends on the context and a brief discussion of the regimes where the defined correctness is satisfactory would strengthen the paper. The example in lines 344 - 355 shows the benefit of treating the function of a program as PAP. This could be balanced by an example showing the drawback (e.g. gradient descent through a non-differentiable point, evaluation of Hamiltonian trajectories for MCMC) of evaluating non-differentiable functions -- even when differentiability a.e. holds.

Correctness: The claims and proofs seem correct.

Clarity: Overall, the paper is well written and pedagogically sound. It is easy to follow and we always know where we are going. The proofs are accessible and the authors do a good job conveying the underlying intuition. Linking the theory back to autodiff systems at the end is a nice touch.

Relation to Prior Work: Yes. The authors take the time to describe existing work and compare it to their own work. Putting this section at the end of the paper, after defining all the relevant terms, is a good idea.

Reproducibility: Yes

Additional Feedback: Going back to paragraph 344 - 355. What is the underlying code for relu, reciprocal_non_nan, and sqrt? It seems the observed behavior is driven by IF statements in the code, a unifying feature that could be highlighted. Optionally, the authors can draw a connection between TAP and generated expression graphs. === After reading rebuttal === I'm very satisfied with the authors' response. What's below constitutes suggestions. I think the point "correctness of the autodiff system (as defined here) doesn't imply correctness of the algorithm" is very important. Gradient descent and HMC are good examples. There are many versions of HMC, and while the one referenced by the authors works well with PAP, others do not. For certain probabilistic programing languages, autodiff is not robust, because it persists in returning a numerical derivative and thence breaking the algorithm. I really would emphasize this point (as seems to be the authors' intention). (For what it's worth, this can motivate other definitions of correctness). > This suggests that “derivatives” of primitive functions f in autodiff systems could have nothing to do with the implementation of f Indeed. The functions here are treated as single nodes on the expression graph, rather than composition of nodes through which autodiff is run. This is not uncommon in autodiff systems, even for more sophisticated functions (matrix operations, numerical solvers,...). The observed behavior is then not so much a property of autodiff, rather of the particular implementation in TensorFlow. The authors could discuss a "hand-coded" relu, which is transcribed on the autodiff stack as a composition of nodes, with the IF statement determining which graph gets generated.

[Author Response · NeurIPS 2020]

We thank the reviewers for their constructive and inspiring feedback. We will improve the paper by incorporating the
following responses. As we cannot see R2 (i.e., Reviewer #2), we respond to the reviews by R1, R3, and R4 only.

**[R1/R4]** The correctness of autodiff systems defined in the paper could be misleading to practitioners. Any relationship
between the correctness of autodiff systems and that of applications (e.g., gradient descent) built upon autodiff systems?

We agree with the reviewers' points that (i) the correctness of the applications built upon autodiff systems is as important
as that of the underlying autodiff systems, but (ii) the former does not necessarily follow from the latter (especially
as defined in the paper). These two notions of correctness address separate issues, and our work is mainly about the
second notion (i.e., the correctness of autodiff systems). Also, we do not claim that our correctness condition is "the"
gold standard. Rather we are just suggesting "a" correctness condition that can serve as a reasonable (possibly minimal)
requirement for existing and future autodiff systems. We will clarify this limitation in the revised version of the paper.

Here are detailed responses to the point (ii) on the applications mentioned in the reviews.

• Gradient descent: As illustrated by R1's example of $f(x)$, our correctness condition for autodiff systems does not
necessarily imply the correctness of the gradient descent based on those systems (i.e., that the gradient descent
converges to Clarke critical points). This gives a partial answer to R3's question on possible drawbacks of using
intensional derivatives. On the other hand, we conjecture that if gradient descent is based on intensional derivatives
and starts at randomly chosen initial points, it would be correct almost surely. This is an open problem.
• Hamiltonian Monte Carlo (HMC) and variational inference (VI), possibly for probabilistic programming: We
reiterate that PAP functions enjoy a nice property that they are analytic on each piece of domain, whose boundary is
measure-zero. The property has been crucially used to design various methods of HMC and VI for non-differentiable
densities and prove their correctness (e.g., [1, 2]). This signifies the importance of studying PAP functions. Whether
the correctness claims in those works would still hold if intensional derivatives are used in place of standard ones, is
another open problem. We will discuss these interesting open problems in the revised version of the paper.

Although there remain a few open problems, we strongly believe that our work would serve as an important first step
towards understanding and resolving those problems. Above all, as far as we know, this is the first work that (i) raises
subtleties in the well-known chain rule when applied to almost-everywhere differentiable functions, (ii) gives concrete
counterexamples illuminating those subtleties, and (iii) proves some reasonable (possibly minimal) correctness of
existing autodiff systems that permits non-differentiable functions, using only elementary mathematics.

[1] Discontinuous Hamiltonian Monte Carlo for discrete parameters and discontinuous likelihoods. Biometrika, 2020.
[2] Reparameterization Gradient for Non-Differentiable Models. In NeurIPS, 2018.

**[R1]** Is there any better correctness condition for autodiff systems that permits some non-differentiable functions but, at
the same time, ensures nice behaviors of autodiff systems when applied to differentiable functions?

This is a good question that would lead to interesting future work. We do not have concrete results, but a possible
approach would be to divide PAP representations and intensional derivatives into "good" and "bad" ones, and consider
a new correctness condition that involves only those "good" ones. A desired property of "good" intensional derivatives
might be that they must be identical to the standard derivative if a given function is differentiable everywhere. Under the
property, R1's $f(x)$ is considered a "bad" PAP representation. A promising idea to construct "good" PAP representations
that induce "good" intensional derivatives is to put additional requirements to their domain partitions, such as that each
piece of a domain partition should be a half-open interval if the entire domain is $\mathbb{R}$; R1's $f(x)$ violates this requirement.

**[R3]** Plots of functions used in the proof of Proposition 1?

Shown right are draft plots of the 1-Cantor function, and $f$ and $g$ in
the proof of Proposition 1. We will add them to the paper.

**[R4]** Implementation of relu, reciprocal_no_nan, and sqrt? Any connections to observed values of their "derivatives"?

We checked that TensorFlow and PyTorch compute the "derivatives" of the three functions $f$, not by applying autodiff
(or symbolic differentiation) to the implementation of $f$, but by evaluating a separately written implementation of $\widetilde{D}f$.
The implementations of $f$ and $\widetilde{D}f$, denoted by `f` and $\widetilde{D}$`f`, are as follows.

| | `relu`$(x)$ | $\widetilde{D}$`relu`$(x)$ | `recip`$(x)$ | $\widetilde{D}$`recip`$(x)$ | `sqrt`$(x)$ | $\widetilde{D}$`sqrt`$(x)$ |
|---|---|---|---|---|---|---|
| 46 TensorFlow | `_max`$(x,0)$ | `if` $(x > 0)$ 1 0 | `_div`$(1,x)$ | `_div`$(-1, x \times x)$ | `_sqrt`$(x)$ | $0.5/$`_sqrt`$(x)$ |
| PyTorch | `if` $(x \le 0)$ 0 $x$ | `if` $(x \le 0)$ 0 1 | N/A | N/A | `_sqrt`$(x)$ | $1/(2 \times$`_sqrt`$(x))$ |

Here `_max` and `_sqrt` are functions provided by a standard math library, and `_div`$(x_1, x_2)$ is implemented as `if` $(x_2 =$
$0)$ 0 $(x_1/x_2)$. If we interpret `relu` and $\widetilde{D}$`relu` as PAP representations, one can see that $(\widetilde{D}$`relu`$) = D($`relu`$)$ may
fail depending on the implementation of `_max` (e.g., consider `_max`$(x_1, x_2) = $`if` $(x_1 \ge x_2)$ $x_1$ $x_2$). This suggests that
"derivatives" of primitive functions $f$ in autodiff systems could have nothing to do with the implementation of $f$.

[Meta-Review · NeurIPS 2020]

The reviewers are unanimous in their appreciation of the theoretical contributions of the paper, while they (and the authors) also agree that the practical implications for applications built on autodiff systems are less well explored. As the authors say, this is an important first step, so is valuable to convey to the NeurIPS audience. The plots proposed by R3 are indeed helpful, so we welcome the authors' proposal to include them in the final copy.